# Evaluating hippocampal replay without a ground truth

**Masahiro Takigawa\*, Marta Huelin Gorriz, Margot Tirole, Daniel Bendor\***

Institute of Behavioural Neuroscience (IBN), University College London (UCL), London, United Kingdom

**Abstract** During rest and sleep, memory traces replay in the brain. The dialogue between brain regions during replay is thought to stabilize labile memory traces for long-term storage. However, because replay is an internally driven, spontaneous phenomenon, it does not have a ground truth - an external reference that can validate whether a memory has truly been replayed. Instead, replay detection is based on the similarity between the sequential neural activity comprising the replay event and the corresponding template of neural activity generated during active locomotion. If the statistical likelihood of observing such a match by chance is sufficiently low, the candidate replay event is inferred to be replaying that specific memory. However, without the ability to evaluate whether replay detection methods are successfully detecting true events and correctly rejecting non-events, the evaluation and comparison of different replay methods is challenging. To circumvent this problem, we present a new framework for evaluating replay, tested using hippocampal neural recordings from rats exploring two novel linear tracks. Using this two-track paradigm, our framework selects replay events based on their temporal fidelity (sequence-based detection), and evaluates the detection performance using each event's track discriminability, where sequenceless decoding across both tracks is used to quantify whether the track replaying is also the most likely track being reactivated.

## Editor's evaluation

In this valuable work, the authors present a case for a new standard in replay detection, tackling the formidable problem that different methods can produce in vastly different results. The authors show compelling evidence about the source of this problem (which is that the true false positive rate can vary wildly between methods). The authors present a solution to the challenge of underestimation of the false positive rate and leverage new experimental data and novel analysis techniques to provide solid evidence that – under specific assumptions – their approach is effective.

**\*For correspondence:**
Masahiro.takigawa.17@ucl.ac.uk (MT);
d.bendor@ucl.ac.uk (DB)

**Competing interest:** The authors declare that no competing interests exist.

## Introduction

The hippocampus plays a central role in the encoding and consolidation of new memories (*Eichenbaum, 2000*; *Frankland and Bontempi, 2005*; *Klinzing et al., 2019*; *Lewis and Bendor, 2019*; *Squire, 1992*). During locomotion in rodents, hippocampal place cells are active in specific regions of the animal's environment (place fields), resulting in a sequential pattern of place cell firing when the animal runs along a spatial trajectory (*O'Keefe and Dostrovsky, 1971*). During offline states, such as quiet restfulness and non-REM sleep, the sequential pattern of neural activity observed during behavior is spontaneously reactivated, a phenomenon referred to as 'hippocampal replay' (*Lee and Wilson, 2002*; *Wilson and McNaughton, 1994*). Offline replay of a neural memory trace is postulated to be a central mechanism by which memories recently encoded in the hippocampus can be consolidated by

further stabilizing these memories in distributed cortico-hippocampal circuits for long-term storage (*Buzsáki, 1989*; *Klinzing et al., 2019*).

Evidence of replay was first discovered almost 30 years ago with the demonstration that during sleep, temporal correlations can be observed between the co-firing of place cell pairs with overlapping place fields (*Wilson and McNaughton, 1994*). Since then, methods for large-scale chronic extracellular neural recordings have become more advanced, allowing the simultaneous recording of tens and even hundreds of neurons (*Ji and Wilson, 2007*; *Lee and Wilson, 2002*; *Pfeiffer and Foster, 2013*; *Widloski and Foster, 2022*). In line with these developments, the analysis methods for replay have also become more sophisticated - shifting from the pairwise analysis of place cells to detecting sequential patterns within neuronal ensembles, using either spiking activity directly (*Foster and Wilson, 2006*) or decoding this activity to extrapolate the virtual spatial trajectories replaying (*Davidson et al., 2009*; *Zhang et al., 1998*). Commonly used replay scoring metrics for quantifying the fidelity of a replay sequence include: (1) a *Spearman's rank-order correlation* of spike times, which quantifies the ordinal relationship between the temporal order of place cell firing during behavior and a replay event's spike train (*Foster and Wilson, 2006*), but assumes that the place cell sequence is ordered accordingly to each place cell's peak firing rate location alone, (2) a *weighted correlation* of the decoded replay event, which quantifies a generalized linear correlation in time and position weighted by the decoded posterior probabilities without any assumption about the temporal rigidity of the replayed trajectory (*Grosmark and Buzsáki, 2016*; *Silva et al., 2015*; *Tirole et al., 2022*), and (3) a *linear fitting* of the decoded replay event, which finds the linear path with the maximum summed decoded probability, assuming that the trajectory's slope is constant (*Davidson et al., 2009*; *Gomperts et al., 2015*; *Ólafsdóttir et al., 2017*). However, because replay is generated by an internal and spontaneous state of the brain, there is no external reference to indicate whether a given replay event is truly a reinstatement of a memory trace. Without a ground truth, the detection of replay events must be inferred based on whether the statistical likelihood of observing a match between the sequential structure of the replayed event and the original behavioral template is sufficiently low, by chance. To quantify this, each event's replay score is compared to a distribution of scores obtained using randomized data, permutated in either the spatial or temporal domain (i.e. a shuffled distribution), where statistically significant replay scores must be greater than a certain percentage of this shuffled distribution, typically 95% for an alpha level ($\alpha$)<0.05.

While recent replay studies have relied predominately on these three major methods of scoring replay (*rank-order*, *weighted correlation*, or *linear fit*), there are still many variations in how these scores can be calculated and how the subsequent statistical significance of these scores are measured. This can lead to issues in reproducibility and a greater difficulty in interpreting conflicting results between studies (*Tingley and Peyrache, 2020*). To overcome this problem, we need the ability to *cross-check* replay events (i.e. are the real events being correctly detected, and non-events being correctly rejected), in spite of not having a ground truth (*Tingley and Peyrache, 2020*; *van der Meer et al., 2017*; *van der Meer et al., 2020*). Hypothetically, any method of cross-checking should at the very least be able to pass a basic test of distinguishing between replay events detected from real data and spurious replay events detected from randomized data.

Given that we are detecting a replay sequence using a replay score (and its significance level $\alpha$), we cannot use similar metrics for cross-checking due to a lack of independence. Here, we solve this problem by evaluating replay sequence events using a sequenceless decoding approach, which has been used in several recent replay studies when the rat's behavior involved running different trajectories, either on a T-maze with two arms or on two different linear tracks (*Carey et al., 2019*; *Tirole et al., 2022*). This framework quantifies how well the sequence-based replay detector discriminates between two track-specific sequences, with a greater difference in summed trajectory likelihoods indicating a higher discriminability.

Several underlying assumptions are required for this framework: (1) for a given replay detection method, the proportion of spurious events (generated from randomized data) that are detected as significant events can be used to empirically estimate the proportion of non-replay events that are falsely labeled as significant replay events in real hippocampal data. However, this assumes that the null distribution used for creating randomized data is not used for detection, and that it does not underestimate the false-positive rates compared to the other null distribution(s) used for detection, (2) for neural data, pooled from multiple animals, with track-specific replay sequences but not spurious

replay events, track discriminability should correlate with the sequenceness score of the track-specific replay, and (3) the empirically estimated false-positive rate can vary between methods, but can be adjusted by a scaling correction of the alpha level such that the performance of different replay methods can be evaluated and compared at an equivalent empirically estimated false-positive rate. Note that this method of scaling the alpha level is designed for method comparison only and should not be meant as a substitute for using appropriate shuffle methods to create a sufficient set of null distributions for detection.

Based on replay data pooled from five rats running on two novel tracks, we validate this framework and demonstrate how measures of sequence fidelity combined with measures of sequenceless track discriminability provide the means to evaluate and compare the performance of different sequence-based replay detection strategies and methods.

## Results

Extracellular signals from the dorsal CA1 region of the hippocampus were recorded in rats running back and forth on two novel linear tracks (male Lister-hooded, *n*=5, 10 sessions, dataset from *Tirole et al., 2022*). In addition to running on two linear tracks (RUN), rats also had a rest/sleep session in a remote location both before (PRE) and afterward (POST) (*Figure 1A*). To demonstrate how this framework quantifies replay detection performance in terms of dataset-specific detection rate, false-positive rate, and track discriminability, we started by detecting replay events using the weighted correlation of the decoded event, a common sequence-based replay detection approach (*Grosmark and Buzsáki, 2016*).

Candidate replay events were population burst events (peak z-scored multi-unit activity [MUA]>3) that occurred during periods of inactivity, when the animal's velocity is less than 5 cm/s (see Methods). Across 10 sessions, we detected 8485, 4643, and 14,326 candidate events in PRE, RUN, and POST session, respectively. Detected replay events were required to also have a ripple power z-score greater than 3 and a significant replay score - i.e., higher than 95% of the scores obtained from each of two shuffle distributions, namely a place field circular shuffle and a time bin permutation shuffle (see Methods) (*Grosmark and Buzsáki, 2016*). The replay score was obtained by performing a weighted correlation on the posterior probabilities (across place and time) obtained using a naïve Bayesian decoder (*Figure 1B and C*). Because the dataset includes more than one track, the posterior probabilities across the two available tracks were normalized at each time bin such that their combined sum was one (*Figure 1B*; *Bendor and Wilson, 2012*; *Carey et al., 2019*).

### Demonstrating a novel framework for cross-checking replay detection performance

We next developed a framework for evaluating and comparing replay detection methods, using a comparison between a replay event's sequence fidelity and its track discriminability. For each replay event detected using a sequence-based detection approach, we also quantified the log odds of reactivation between tracks, a sequenceless metric *based on only place cells with place fields on both tracks*, to avoid any potential bias in track discrimination (*Figure 1D and E*, see Methods) (*Carey et al., 2019*; *Tirole et al., 2022*). Sequenceless decoding involved three steps: (1) computing the summed posteriors (across time and space) within the replay event for each track, (2) calculating log of the ratio between the summed posteriors for each track, and (3) taking the z-score of this value, based on a distribution of log odds computed by a track ID shuffle (*Carey et al., 2019*; *Tirole et al., 2022*). For each place cell in the track ID shuffle, the corresponding track 1 and 2 place fields were randomly assigned to the correct track or swapped. As a result, a more positive z-scored log odds would indicate a greater likelihood of track 1 reactivation whereas a more negative value would indicate a greater likelihood of track 2 reactivation (*Figure 1E* and *Figure 1—figure supplement 1*). To use sequenceless decoding to evaluate replay events, we computed the *difference* in mean log odds between track 1 and track 2 replay events across all sessions, which were originally detected using standard sequence-based replay detection methods. For this measurement, a more positive mean log odds difference would indicate a higher track discriminability in the replay content, and a higher confidence in replay quality. In contrast, a mean log odds difference of 0 would suggest that the quality of

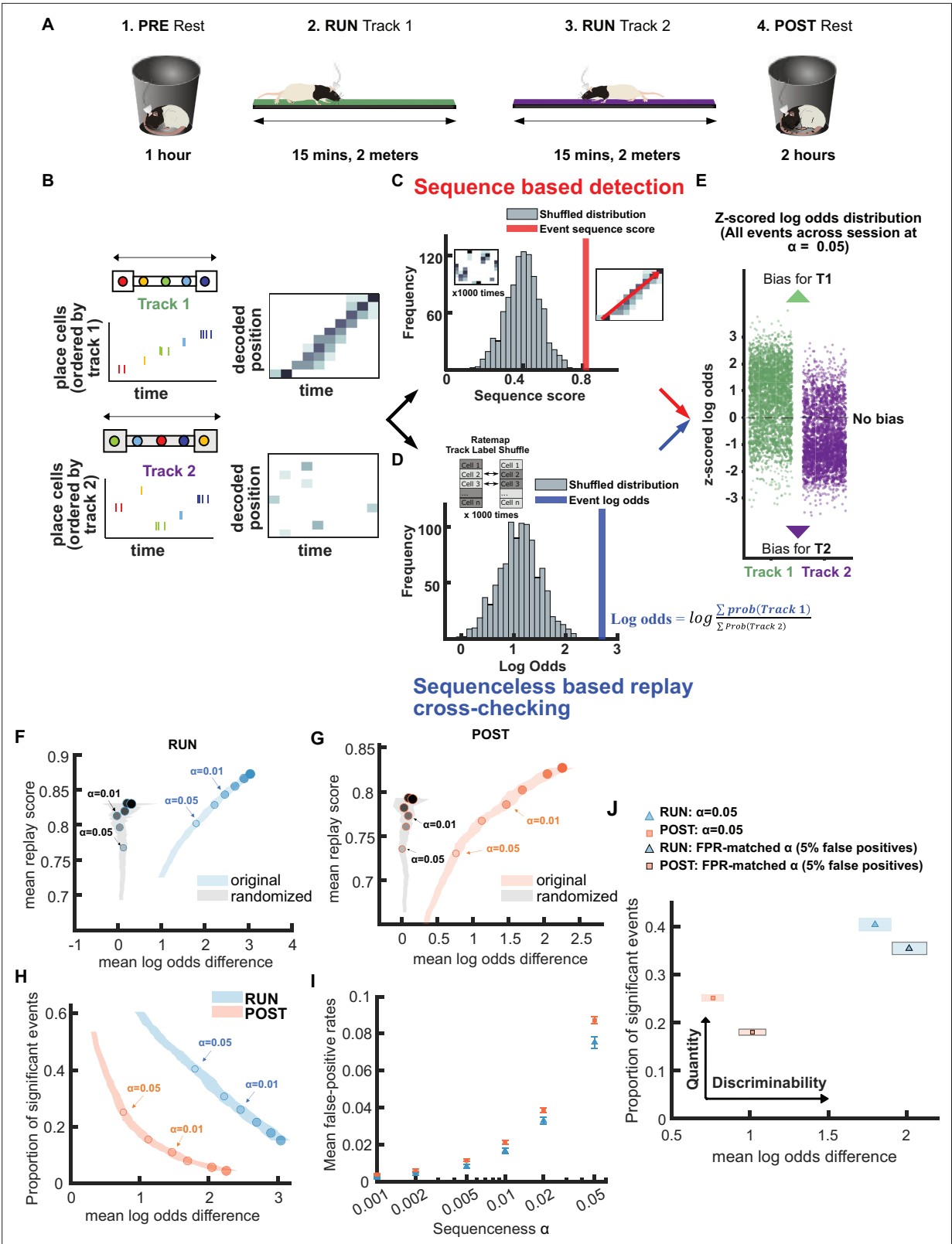

**Figure 1.** Demonstration of novel replay analysis framework for comparing sequence fidelity with track discriminability. (**A**) Experimental design. For each recording session, the animal ran back and forth on two novel linear tracks (RUN) with resting sessions before (PRE) and afterward (POST). Rat schematic in **A** was adapted with permission from SciDraw.io (*Asiminas, 2020b*; *Asiminas, 2020a*). (**B–E**) Schematic of sequence-based and sequenceless decoding framework. (**B**) Each candidate replay event spike train was fed into a naïve Bayesian decoder to calculate the decoded

*Figure 1 continued*

posterior probabilities across time and space. (**C**) Then, for the sequence-based analysis, the sequence score for each candidate event was determined from the weighted correlation of the posterior probability matrix. Significance was determined by comparing the replay score relative to a shuffled distribution (alpha level = 0.05). (**D**) For sequenceless decoding, a Bayesian decoding similar to sequence-based approach was used, with the exception that only place cells with stable place fields on both tracks were used as template to avoid any track discrimination bias. Then, the logarithmic ratio of the summed posterior probabilities within each replay event for each track is calculated (log odds). The event log odds were z-scored relative to a shuffled distribution where each place cell's track 1 and 2 place fields were randomly shuffled between tracks. (**E**) The difference between track 1 and track 2 replay events' log odds can be used as a metric to cross-check the performance of sequence-based replay detection. (**F,G**) The relationship between the mean log odds difference and mean replay score for the significant events detected at different alpha levels (0.2–0.001) using a weighted correlation replay scoring with two different shuffling procedures (place field circular shuffle and time bin permutation shuffle). The shaded region indicates the 95% bootstrap confidence interval for the mean log odds difference. The six dots with increasing color intensity for each distribution represent the data at an alpha level of 0.05, 0.02, 0.01, 0.005, 0.002, and 0.001. (**F**) Significant events detected during RUN using original candidate events (blue) and cell-id randomized spurious events (gray). (**G**) Significant events detected during POST using original candidate events (orange) and cell-id randomized spurious events (gray). (**H**) The relationship between the mean log odds difference and the proportion of significant events detected at different alpha levels (0.2–0.001) during RUN (blue) and POST (orange).The shaded region indicates the 95% bootstrap confidence interval for mean log odds difference. The six dots with increasing color intensity for each distribution represent the data at an alpha level of 0.05, 0.02, 0.01, 0.005, 0.002, and 0.001. (**I**) The mean proportion of significant cell-id randomized events (mean false-positive rate) at different alpha levels (0.2–0.001). The error bar indicates the 95% bootstrap confidence interval for mean false-positive rates. (**J**) The replay detection performance at the original alpha level = 0.05 and the FPR-matched alpha level when the mean false-positive rate was 5%. The shaded box indicates a 95% bootstrap confidence interval for both proportion of significant events detected and mean log odds difference. The box with a light outline represents the values at an alpha level = 0.05 and the box with black outline represents the values at the FPR-matched alpha level. (Number of candidate replay events: RUN n = 4643 and POST n = 15283). The 95% confidence interval for the proportion of significant events, mean log odds difference, mean false-positive rates, and the FPR-matched alpha level for replay events detected during RUN and POST are available in *Figure 1—source data 1*.

The online version of this article includes the following source data and figure supplement(s) for figure 1:

**Source data 1.** Summary of replay detection performance for weighted correlation method with two shuffles.

**Figure supplement 1.** Individual examples of decoded trajectories with different sequence fidelity and reactivation bias.

**Figure supplement 2.** Mean log odds difference and proportion of significant events detected hold across 10 sessions.

**Figure supplement 3.** Schematics of shuffling procedures performed to obtain four null distributions for replay detection.

**Figure supplement 4.** Proportion of false-positive events detected when using different shuffle methods and randomized datasets for a null distribution.

replay events is comparable to chance level track discriminability, and that the detected event population most likely consists of low fidelity events, indistinguishable from false-positives.

As mentioned earlier, any method of cross-checking should at least be able to pass a basic test of distinguishing between replay events detected from real data and spurious replay events detected from randomized data. We first compared the mean log-odds difference and mean weighted correlation score for replay events detected using our dataset and spurious replay events detected after the same dataset was randomized using a cell-id shuffle. Note that all sequence-based replay detection methods in this study did not use a cell-id shuffle, and as such this shuffling approach was sufficiently independent for generating random sequences as a negative control. When the alpha level, the significance level for detecting replay events, was varied from 0.2 to 0.001, we observed a close relationship between mean weighted correlation score and alpha level for both the original and randomized dataset (*Figure 1F and G*). However, there was a clear dissociation between the alpha level and mean log odds difference when using a randomized dataset, for both RUN and POST replay events. This demonstrated that while the weighted correlation score used for replay detection improved with alpha level even for the randomized dataset, the mean log odds difference (track discriminability) was able to still differentiate real from spurious replay events, acting independently from our selection criteria of replay events based on sequence fidelity.

While an alpha level of 0.05 is usually the chosen threshold for whether a statistical test rejects the null hypothesis, we predicted that as the alpha level became stricter, the mean log odds difference should increase due to a lower rate of false-positive events, albeit at the cost of also fewer detected replay events overall. To study the trade-off between the track discriminability and detection rate of detected replay events, we first analyzed our POST and RUN replay events, comparing the number of significant events detected on both tracks to the mean log odds difference, as the alpha level was adjusted between 0.2 and 0.001 (*Figure 1H*). For both POST and RUN, the mean log odds difference increased (higher track discriminability) as the alpha level threshold decreased, which also

corresponded to a decrease in the number of detected events. In addition, we observed RUN replay events to be comparatively more prevalent and yielding a higher track discriminability than POST replay events, in line with previous reports (*Karlsson and Frank, 2009*; *Tirole et al., 2022*). This might be partly due to the presence of immediate sensory or other external inputs helping to direct hippo-campal place cell ensembles toward representing the local current environment during awake replay.

To determine if this decrease in the detection rate was also associated with a decrease in the false-positive rate, we empirically measured the mean fraction of spurious replay events detected across both tracks after the dataset was randomized (by permuting the cell-id of each place cell) as a proxy for false-positive rate. We observed that as the alpha level was gradually reduced, so was the estimated false-positive rate using cell-id randomization (*Figure 1I*). We also found that the estimated false-positive rate was higher than the alpha level: using an alpha level of 0.05, 7.5% (lower CI = 7.2% and higher CI = 7.8%) of RUN replay events were false-positives, while 8.7% (lower CI = 8.5% and higher CI = 8.9%) of POST replay events were false-positives. Using this information, we could quantify the proportion of significant events for both tracks using sequence-based detection, and the corresponding mean log odds difference at the original alpha level of 0.05 and an FPR (false-positive rate)-matched alpha level (RUN: $\alpha = 0.032$, POST: $\alpha = 0.028$) that adjusted the estimated mean false-positive rate to approximately 5% (*Figure 1J*). These results were generally consistent when applied to individual sessions (*Figure 1—figure supplement 2*). Such adjustment of the alpha level based on a matching false-positive rate is an important part of this novel framework as it provides an opportunity to compare different replay detection strategies and methods in a more equitable manner.

However, since most replay detection methods rely on comparing each event's sequence score to one or more null distributions, it is important to make sure that cell-id randomization was not underestimating the false-positive rate of replay detection in a way that was biased toward specific null distributions used for detection. Therefore, we examined the influence of data randomization on empirically estimated false-positive rates by comparing four sets of randomized data: (1) cell-id randomization, (2) spike train circular shifted dataset, (3) place field circular shifted dataset, and (4) a cross-experiment shuffled dataset (where place fields from a different recording session were randomly assigned to each place cell during Bayesian decoding). We next performed replay detection using a weighted correlation with four single shuffle methods including two pre-decoding shuffles (spike train circular shuffle, place field circular shuffle) and two post-decoding shuffles (place bin circular shuffle, time bin permutation shuffle) (*Figure 1—figure supplement 3*). In most cases, both the cell-id randomized dataset and cross-experiment-shuffled dataset led to highest empirically estimated false-positive rates (*Figure 1—figure supplement 4*). However, using a shuffling procedure during detection that randomized the data along the same dimension (time or place) as the randomized dataset led to a substantial underestimation of the empirically estimated false-positive rate (*Figure 1—figure supplement 4*). Therefore, the cell-id randomization approach (and alternatively the cross-experiment shuffling approach) were the more optimal methods to create a randomized dataset for empirically estimating false-positive rates as the null distribution was not directly used in replay detection and was not underestimating the false-positive rates compared to alternative methods of data randomization.

We next used our framework to analyze and compare different strategies for replay detection. We did not attempt to precisely replicate any specific published replay detection method, as our goal was to see how general methodological differences can impact the track discriminability and quantity of detected events, and whether there is a preferred general approach for replay analysis. While it is important to note that there are many subtler, data-specific details to potentially consider (e.g. number of cells, smoothing of data, bin size, etc.) that can impact both the estimated false-positive rate and quality of detected replay events, our aim was to provide an overarching framework to help identify the more optimal replay detection approach.

## Ripple power is an important criterion for replay event selection during POST

During both quiet wakefulness and non-REM sleep, replay events preferentially occur during sharp-wave ripple events, transient high-frequency oscillations (150–250 Hz) in the local field potential (LFP) recorded near the cell layer of CA1 (*Buzsáki, 2015*; *Foster and Wilson, 2006*; *Lee and Wilson, 2002*). As such, a minimum ripple power has been used as a criterion for detecting candidate replay events prior to sequence detection, however the threshold has varied substantially across previous

studies ranging from 2 SD (standard deviations) above baseline (*Diba and Buzsáki, 2007*; *Gillespie et al., 2021*) to 7 SD above baseline (*Csicsvari et al., 2007*; *Ji and Wilson, 2007*). However, because many replay events consist of the spontaneous reactivation of a large proportion of place cells within a short time window (more than typically occurs during active behavior), a substantial increase in MUA during immobility also provides a reasonable criterion for detecting candidate replay events (*Davidson et al., 2009*). Because using only MUA as a threshold typically leads to more candidate replay events, and is believed to be more reliable for detecting both the start and end times of the replay event, many recent studies have opted to no longer use minimum ripple power as a criterion for candidate replay events (*Gomperts et al., 2015*; *Ólafsdóttir et al., 2018*; *Ólafsdóttir et al., 2016*; *Silva et al., 2015*).

We next applied our replay analysis framework to test how a stricter criterion for ripple power affects track discriminability. Candidate events were selected based on both elevated MUA (z-score>3) and ripple power *limited to a specific range*, measured in SD above baseline (i.e. a z-score of 0–3, 3–5, 5–10, or >10). For equivalent alpha levels, a stricter ripple threshold resulted in both a higher proportion of significant events and a higher mean log odds difference (*Figure 2A and B*). The increase in track discriminability observed with higher ripple power was not associated with a change in the number of active place cells or total place cell spiking activity (*Figure 2—figure supplement 1*). However, one major difference between RUN and POST replay events was that even at the lowest ripple power threshold, RUN replay events had a non-zero mean log odds difference, which increased with stricter alpha levels. In contrast for POST replay events with a ripple power less than a z-score of 5, the mean log odds difference was near zero for all alpha levels tested. Furthermore, while the proportion of detected POST events with a ripple power less than a z-score of 5 was approximately 20% at an alpha level of 0.05, the proportion of detected POST events dropped to near chance levels of 10% when the alpha level was adjusted to match the mean false-positive rate of 5% (where 10% of spurious events were detected as significant events across both tracks) (*Figure 2C and D* and *Figure 2—figure supplement 2*, see Methods). For ripple thresholds above 5, the log odds difference increased as the proportion of events and the alpha level decreased (*Figure 2A and B*). Similar results were observed if stricter criteria were used in detecting candidate replay events - either lowering the speed threshold criterion (<1 cm/s) or imposing an additional requirement for low theta power (z-score<0) (*Figure 2—figure supplement 3*), suggesting that our finding was not confounded by the presence of other neuronal sequences during movement such as theta sequences. Next, we tested how shifting the ripple threshold (between 0 and 10 SD above baseline) could impact overall detection rates (*Figure 2—figure supplement 4*), calculating the proportion of significant events out of all of the candidate events based on MUA criteria alone. Consistent with our main finding, while the proportion of significant events dropped as a result of ripple thresholding, only replay during POST but not RUN improved its log odds difference with a stricter ripple threshold. These results emphasize that a ripple power threshold is not necessary for RUN replay events in our dataset but may still be beneficial, as long as it does not reject too many good replay events with low ripple power. In other words, depending on the experimental design, it is possible that a stricter alpha level with no ripple threshold can be used to detect more replay events than using a less strict alpha level combined with a strict ripple power threshold. However, for POST replay events, a threshold at least in the range of a z-score of 3–5 is recommended based on our dataset, to reduce inclusion of potential false-positives within the pool of detected replay events. For the remainder of this study, we incorporated a strict, albeit less conservative ripple power threshold (z-score>3) for both RUN and POST, given the trade-off between an improved track discriminability and a decrease in the number of detected events. This also allowed a more direct comparison of awake and rest replay using identical criteria while still being consistent with previous replay studies (*Karlsson and Frank, 2009*; *Pfeiffer and Foster, 2013*; *Todorova and Zugaro, 2019*).

## Replay detection is sensitive to the shuffle method used

Statistical significance of each replay event is calculated by comparing the replay score to one or more Monte-Carlo shuffle distributions, with each shuffle designed to randomize a specific aspect of sequential place cell firing while attempting to keep other factors intact. Some shuffles are applied to randomize the place fields or the spike train of the original data prior to Bayesian decoding, while other shuffles are applied to the spatial or temporal dimension of the decoded posterior probabilities.

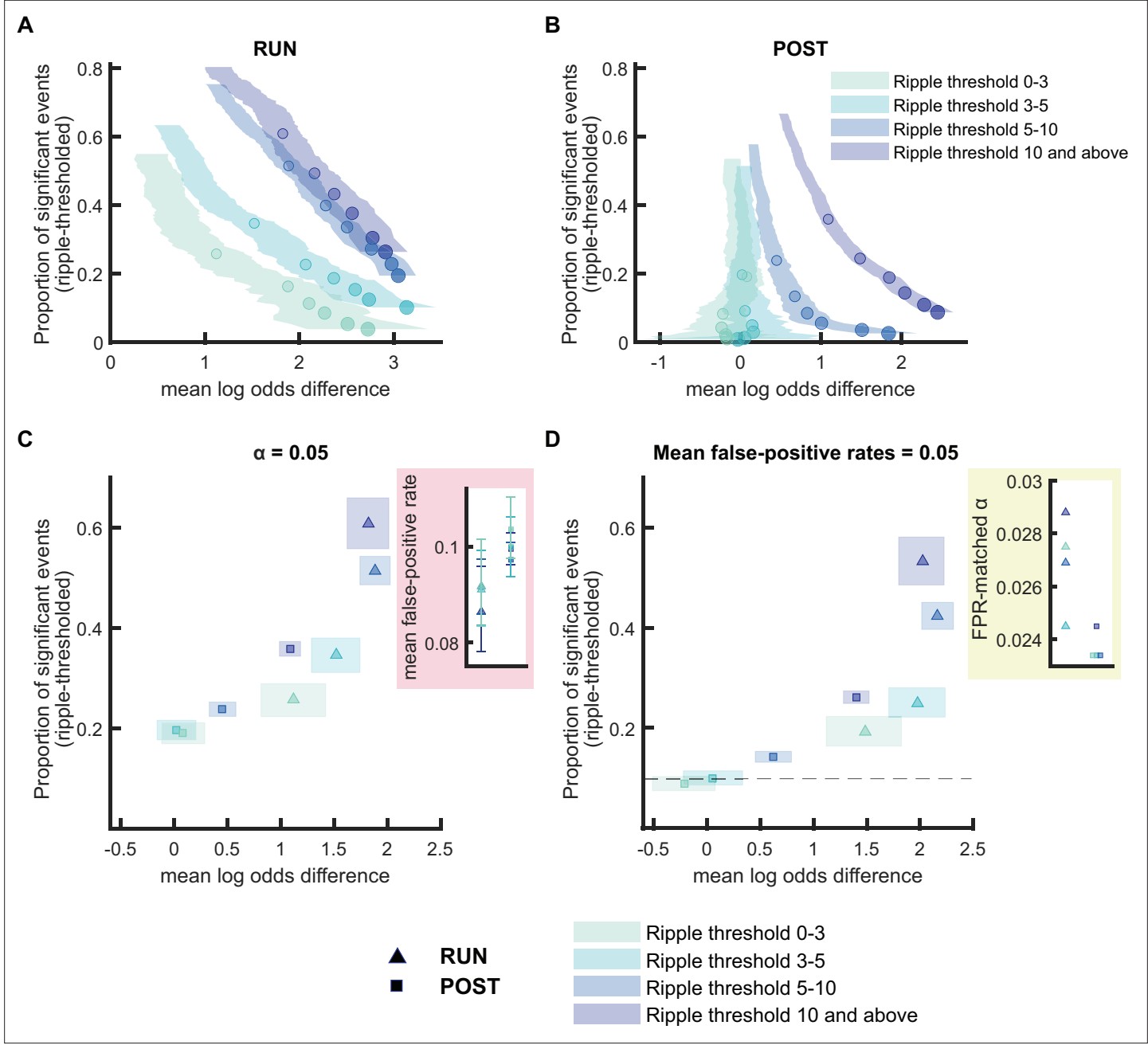

**Figure 2.** Replay detection performance improves with ripple power. (**A,B**) The proportion of significant events and mean log odds difference at different alpha levels (0.2–0.001) as ripple power increases (0–3, 3–5, 5–10, 10, and above). The shaded region indicates the 95% bootstrapped confidence interval for mean log odds difference. The six dots with increasing color intensity for each distribution represent the data at an alpha level of 0.05, 0.02, 0.01, 0.005, 0.002, and 0.001. (**A**) Replay events detected during RUN. (**B**) Replay events detected during POST. (**C,D**) The proportion of significant events and mean log odds differences at (**C**) an alpha level = 0.05 and (**D**) an FPR-matched alpha level with a mean false-positive rate of 5%. The shaded box indicates a 95% bootstrap confidence interval for both the proportion of significant events detected and mean log odds difference. The triangle symbol is used to represent replay events during RUN and the square symbol is used to represent replay events during POST. The dashed line represents the approximate chance level at mean false-positive rate of 5%. (Number of candidate replay events for ripple range 0-3, 3-5, 5-10 and 10 and above: RUN n = 782,1091,1982 and 788 and POST n = 1667, 2136, 4904 and 5619, respectively). The 95% confidence interval for the proportion of significant events, mean log odds difference, mean false-positive rates, and the FPR-matched alpha level for replay events with different ripple power range are available in *Figure 2—source data 1*.

The online version of this article includes the following source data and figure supplement(s) for figure 2:

**Source data 1.** Summary of replay detection performance at different ripple power thresholds.

*Figure 2 continued on next page*

*Figure 2 continued*

**Figure supplement 1.** The replay event distribution, number of active place cells, and total spikes during replay event at different ripple powers.

**Figure supplement 2.** The mean false-positive rate across both tracks for replay events detected with different ripple power range.

**Figure supplement 3.** Replay detection performance improves with ripple power using stricter criteria for candidate events.

**Figure supplement 4.** Replay detection performance improves with ripple threshold for POST but not RUN.

Because place cells often fire bursts of spikes, this creates non-independent samples, which *violate the assumption of having independent samples for a statistical test*. Thus, it is important for a shuffle method to preserve such aspects of the data that are not independent, to avoid adding type 1 errors in the statistical analysis, leading to a false-positive rate that exceeds the alpha level. Given this, we next examined how different shuffling procedures impact replay detection performance. Using the weighted correlation scoring method, we examined four types of shuffling procedures to see if they differed in how they detected replay during RUN and POST epochs. These consisted of two pre-decoding shuffles (spike train circular shuffle, place field circular shuffle) and two post-decoding shuffles (place bin circular shuffle, time bin permutation shuffle) (*Figure 3A–D*, *Figure 3—figure supplement 1*).

For both RUN and POST, we observed that pre-decoding shuffles performed better than post-decoding shuffles, with the time bin permutation shuffle showing a significantly lower log odd difference compared to the other three shuffles (*Figure 3A and B*). Although the post-decoding time bin permutation shuffle seemingly led to the highest detection rate at an alpha level of 0.05 (*Figure 3C*), we observed that its mean false-positive rate at an alpha level of 0.05 was also substantially higher than other shuffling procedures. When we adjusted the alpha level to match a mean false-positive rate of 5%, the post-decoding time bin permutation shuffle had the poorest performance (the lowest proportion of significant events and the lowest mean log odds difference) across the four types of shuffles for both RUN and POST epochs (*Figure 3D*). For both the RUN and POST epochs, the remaining three shuffling procedures had a similar mean log odds difference when using an FPR-matched alpha level. However, while the proportion of significant events were similar across the three shuffling methods for POST, the two pre-decoding shuffling procedures had a higher proportion of significant events for RUN replay events. These results suggested that the shuffling procedures that directly manipulated the place field data or spike train data before decoding may be more efficacious than procedures that randomized the posterior probability distributions after decoding.

## Replay detection can be improved by adding stricter detection criteria

Following our comparison of different shuffling procedures, we next investigated how the inclusion of additional criteria in replay detection could further improve performance. Commonly this is accomplished by using multiple types of shuffling procedures, with the requirement that each shuffling method must independently pass the same alpha level for the replay event to be classified as statistically significant. An alternative approach to using multiple shuffling procedures is to impose a criteria of jump distance - the number of spatial bins the decoded estimated position is allowed to jump across neighboring time bins during a spatial trajectory, with candidate replay events rejected if they fail this criterion (*Silva et al., 2015*). One justification for using this metric is that a lower jump distance is correlated with a higher weighted correlation score, however not all events with higher scores are necessarily higher quality events.

Based on this, we compared the performance of four replay detection methods varying in their shuffling procedures: (1) a single place field circular shuffle, (2) a place field circular shuffle with a jump distance threshold (40% of track) (*Silva et al., 2015*), (3) two shuffles (place field circular shuffle and time bin permutation shuffle) (*Grosmark and Buzsáki, 2016*), and (4) three shuffles (place field circular shuffle, spike train circular shuffle, and place bin circular shuffle) (*Tirole et al., 2022*). For both RUN and POST epochs, increasing the number of shuffles increased the mean log odd difference and decreased the proportion of significant events and the estimated false-positive rate, for similar alpha levels (*Figure 4A and B* and *Figure 4—figure supplement 1A and B*). Adding a jump distance threshold as an additional criterion also increased the mean log odd difference, although this was accompanied by a marked reduction in the proportion of significant replay events detected (*Figure 4A and B*). Before controlling for the estimated false-positive rates associated with each

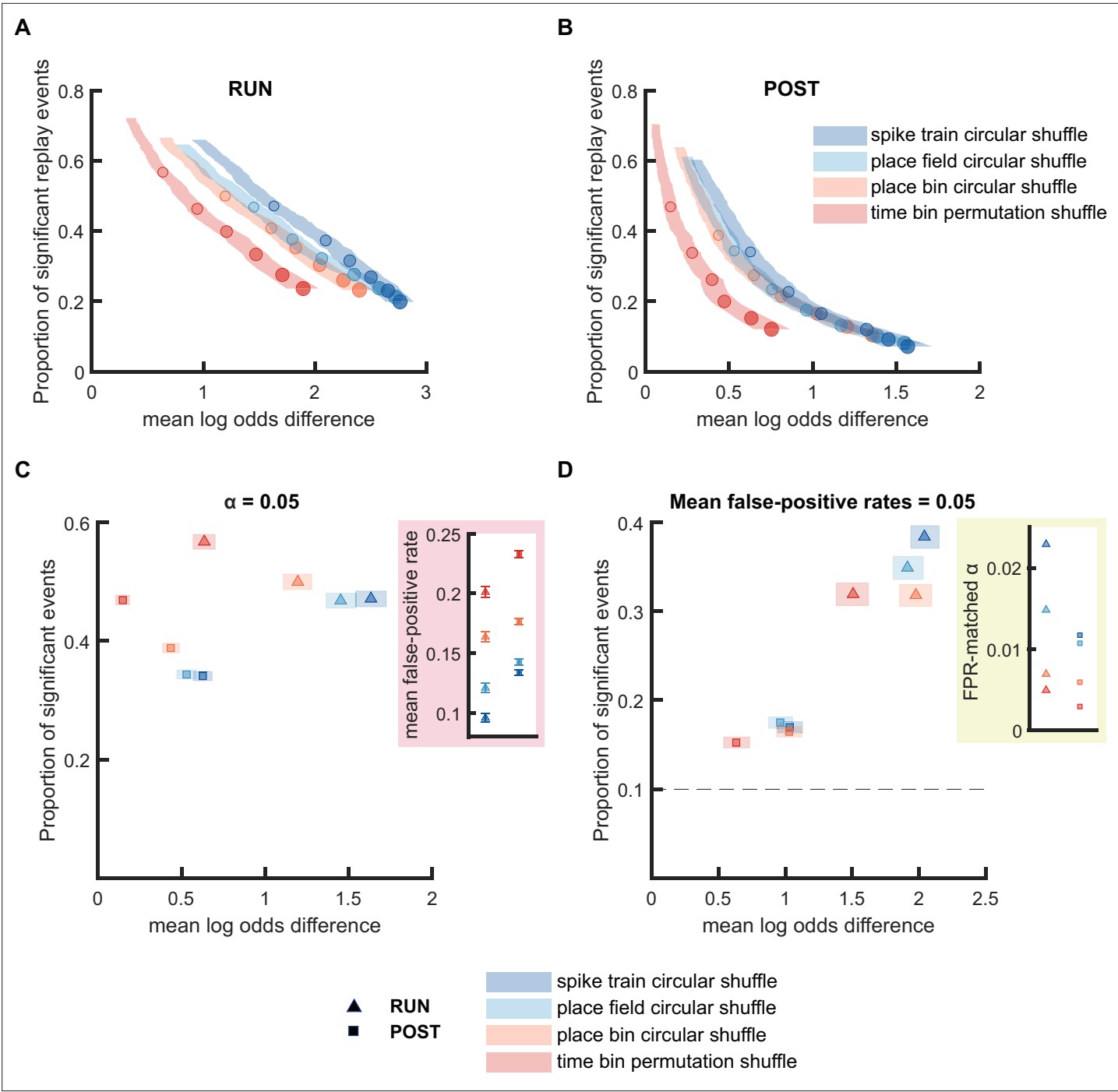

**Figure 3.** Replay detection performance was sensitive to the shuffling method applied. (**A,B**) The proportion of significant events and mean log odds difference at different alpha levels (0.2–0.001) when using four different shuffling methods: (1) spike train circular shuffle (dark blue), (2) place field circular shuffle (light blue), (3) place bin circular shuffle (orange), and (4) time bin permutation shuffle (red). The shaded region indicates the 95% bootstrapped confidence interval for mean log odds difference. The six dots with increasing color intensity for each distribution represent the data at an alpha level of 0.05, 0.02, 0.01, 0.005, 0.002, and 0.001. (**A**) Replay events detected during RUN. (**B**) Replay events detected during POST. (**C,D**) The proportion of significant events and mean log odds difference at (**C**) an alpha level = 0.05 and (**D**) an FPR-matched alpha level with a mean false-positive rate of 5%. The shaded box indicates the 95% bootstrap confidence interval for both the proportion of significant events detected and mean log odds difference. The triangle symbol is used to represent replay events during RUN and the square symbol is used to represent replay events during POST. The dashed line represents the approximate chance level at mean false-positive rate of 5%. (Number of candidate replay events: RUN n = 4643 and POST n = 15283). The 95% confidence interval for the proportion of significant events, mean log odds difference, mean false-positive rates, and the FPR-matched alpha level for replay events detected using different shuffling methods are available in *Figure 3—source data 1*.

*Figure 3 continued on next page*

*Figure 3 continued*

The online version of this article includes the following source data and figure supplement(s) for figure 3:

**Source data 1.** Summary of replay detection performance using four different shuffling methods.

**Figure supplement 1.** The mean false-positive rate across both tracks for replay events detected using different shuffling methods.

method, as expected, the stricter methods would detect fewer replay events than more permissive methods. For instance, at alpha level of 0.05, around 50% of candidate events would be detected as significant events when using single shuffle, as opposed to 40% when using two shuffles (*Figure 4C*). However, after adjusting the alpha level to a matching false-positive rate of 0.05, we found that stricter methods would detect more replay events. For example, at an FPR-matched alpha level, the proportion of significant events would reduce to 32% for a single shuffle ($\alpha$ = 0.007) but 35% for the stricter two-shuffle method ($\alpha$ = 0.032). This shows that controlling for the estimated false-positive rate can, almost paradoxically, result in stricter methods with additional shuffle types detecting more replay events. When using the original alpha level of 0.05, two methods (1 shuffle with a jump distance threshold and 3 shuffles) resulted in the highest mean log odd difference (*Figure 4C*). However, if an FPR-matched alpha level was used (with the estimated false-positive rate fixed at 0.05), only the use of 2 or more shuffles achieved both the highest proportion of events and largest mean log odds difference, for both RUN and POST epochs (*Figure 4D*). In contrast to this, a single shuffle with an added jump distance criterion performed the poorest (less events and lower track discriminability) out of the four methods when using the FPR-matched alpha level. This suggests that it is more beneficial to use a stricter alpha level than to impose a jump distance, an observation that extended to a range of different jump distances spanning 20–60% of the track (*Figure 4—figure supplement 2*).

## Rank-order-based replay detection is sensitive to spike selection

We next extended our framework to investigate other replay scoring methods - first focusing on the use of rank-order correlation. Instead of scoring the decoded trajectories, this approach directly quantifies the ordinal relationship within the spiking patterns (*Foster and Wilson, 2006*). In this method, the sequential order of place fields is ranked along the track, and compared with the corresponding rank of the spike times within the replay event. Rank-order correlation has been previously applied in several ways: (1) to all spikes within the replay event (*Dragoi and Tonegawa, 2011*), (2) to all proto-events, each representing a bursts of spikes (*Foster and Wilson, 2006*), or (3) only to one spike emitted by each place cell during a candidate replay event (e.g. median or first spike) (*Tirole et al., 2022*). However, as mentioned previously, place cells often fire bursts of spikes which in effect creates non-independent samples. Therefore, in theory, the inclusion of all spikes for the Spearman's rank-order based analysis would violate the statistical assumption of independent samples and lead to more type I errors (and a higher false-positive rate). We compared the rank-order-based analysis using either all spikes or median spike in our framework, and found that the estimated false-positive rate when using independent events (median spike time) remained around 5% (lower CI = 5.2% and upper CI = 5.8%), but increased to around 17% (lower CI = 16.3% and upper CI = 17.2%), when using all spikes (*Figure 5* and *Figure 5—figure supplement 1*). However, when using the FPR-matched alpha level (false-positive rate fixed at 5%), the trade-off between the proportion of significant events and their mean log odds difference was similar between the two methods (*Figure 5D* and *Figure 5—figure supplement 1*).

## Replay events have a similar quality across different detection methods, when using an FPR-matched alpha level, but vary in their proportion

Finally, we expanded our analysis to directly compare every replay scoring and shuffling method already implemented in this report, with the addition one more commonly used method - a linear fit-based score of decoded trajectories with either one and two types of shuffles (*Davidson et al., 2009*; *Muessig et al., 2019*; *Ólafsdóttir et al., 2017*). In the linear fit approach, the decoded posterior probability matrix, representing the probability of the virtual trajectory at each position on the track and at each time point, is compared to the best linear trajectory; the corresponding replay score is the sum of all posterior probabilities within a certain distance of this line, after maximizing this

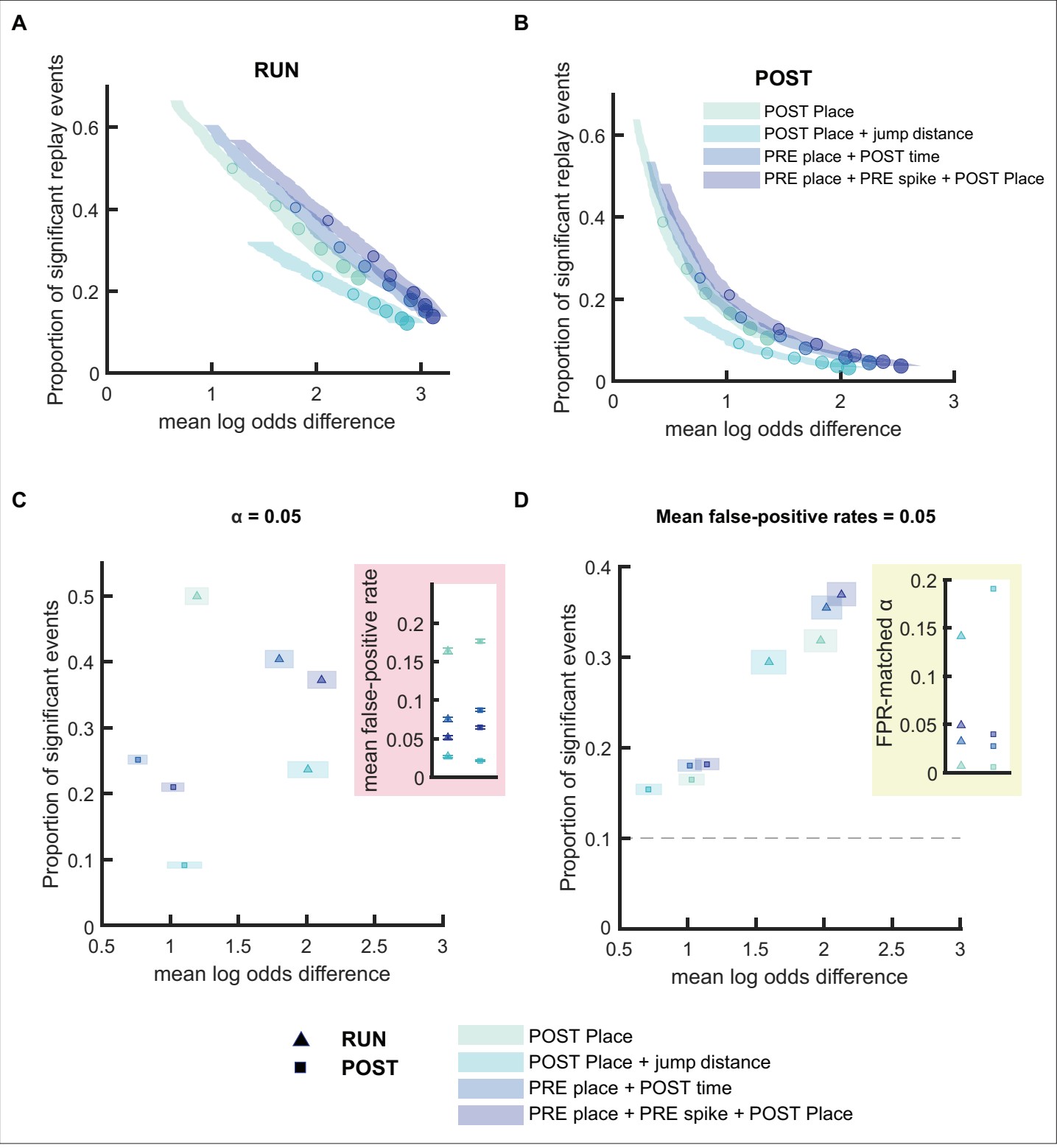

**Figure 4.** Replay detection performance can be improved by adding stricter detection criteria. (**A,B**) The proportion of significant events and mean log odds difference at different alpha levels (0.2–0.001) with four different detection criteria: (1) Only a post-decoding place bin circular shuffle, (2) a post-decoding place bin circular shuffle with jump distance threshold at normalized track length 0.4, (3) a pre-decoding place field circular shuffle and a post-decoding time bin permutation shuffle, (4) a pre-decoding place field circular shuffle, a pre-decoding spike train circular shuffle, and a post-decoding place bin circular shuffle. The shaded region indicates a 95% bootstrap confidence interval for mean log odds difference. The six dots with increasing color intensity for each distribution represent the data at an alpha level of 0.05, 0.02, 0.01, 0.005, 0.002, and 0.001. (**A**) Replay events detected

*Figure 4 continued on next page*

*Figure 4 continued*

during RUN. (**B**) Replay events detected during POST. (**C,D**) The proportion of significant events and mean log odds difference at (**C**) an alpha level = 0.05 and (**D**) an FPR-matched alpha level with a mean false-positive rate of 5%. The shaded box indicates a 95% bootstrap confidence interval for both the proportion of significant events detected and mean log odds difference. The triangle symbol represents replay events during RUN and the square symbol represents replay events during POST. The dashed line represents the approximate chance level at mean false-positive rate of 5%. (Number of candidate replay events: RUN n = 4643 and POST n = 15283). The 95% confidence interval for the proportion of significant events, mean log odds difference, mean false-positive rates, and the alpha level for replay events detected using different detection criteria are available in *Figure 4—source data 1*.

The online version of this article includes the following source data and figure supplement(s) for figure 4:

**Source data 1.** Summary of replay detection performance when adding stricter detection criteria.

**Figure supplement 1.** The mean false-positive rate across both tracks for replay events detected using different detection criteria.

**Figure supplement 2.** Comparison of the replay detection performance when applying different jump distance thresholds.

**Figure supplement 2—source data 1.**

score across all possible lines that can pass through this matrix. While the linear fit approach is more stringent than the weighted correlation, it may also be less sensitive to non-linear replay trajectories (e.g. sigmoidal). When using a cell-id permuted randomized dataset to examine the chance-level detection for each method, we observed no bias in the mean log odds difference for every method, except for a linear fit approach with two shuffles (*Figure 6—figure supplement 1*). To avoid any possible intrinsic bias in our calculation of the mean log odds, we calculated the *adjusted mean log odds difference*, by subtracting any potential bias in log odds difference measured using spurious replay events (*Figure 6*). Similar results were observed using an alternative within-track normalization approach in Bayesian decoding prior to replay detection (*Figure 6—figure supplement 2*). Finally, to provide a more comprehensive comparison across methods, we also included replay events detected in the PRE epochs, a phenomenon referred to as de novo preplay where statistically significant sequences are detected prior to any knowledge or experience running on the track (*Dragoi and Tonegawa, 2011*).

Using our framework to compare replay detection methodologies, which incorporated (1) the corrected mean log odds difference, (2) the empirically estimated mean false-positive rate based on replay detection using a randomized dataset (cell-id permutation), and (3) the proportion of significant events on both tracks, we observed four main findings regarding replay during three different behavioral epochs (*Figure 6* and *Figure 6—figure supplements 3–5*). First, the proportion of significant events varied substantially across methods when using the original alpha level of 0.05, however this was consistent with the estimated false-positive rate, which ranged from 2% to 18%. Moreover, the proportion of mean false-positive rates estimated were also highly correlated with the proportion of events being classified as significant for both tracks (dual-track events), suggesting that most dual-track events we observed were likely due to false-positive detection on one of the two tracks rather than multiple track representations in a single replay event (*Figure 6—figure supplement 6*). Second, using the FPR-matched alpha level, all methods had a proportion of significant preplay events close to 10%, which was similar to the proportion of false-positive events expected by chance for two tracks. Furthermore, the shuffle-subtracted mean log odds difference for preplay was approximately zero for all methods, suggesting that regardless of the method used, if used correctly, preplay is indistinguishable from randomly generated spurious replay events. Third, using the FPR-matched alpha level for POST replay events, all methods produced a corrected mean log odds difference near 1, however linear and weighted correlation methods with 2 or more shuffles detected a higher proportion of significant events. Finally, using the FPR-matched alpha level for RUN replay events, the majority of methods produced a corrected mean log odds difference near 2. Despite using the FPR-matched alpha level to achieve the same mean false-positive rate, we observed that using two or more shuffling procedures would lead to significant improvement in the overall replay detection performance for both weighted correlation and linear fitting method. Replay detection methods will continue to evolve and be refined in the future; the quantification of any improvements is now possible with the framework outlined here.

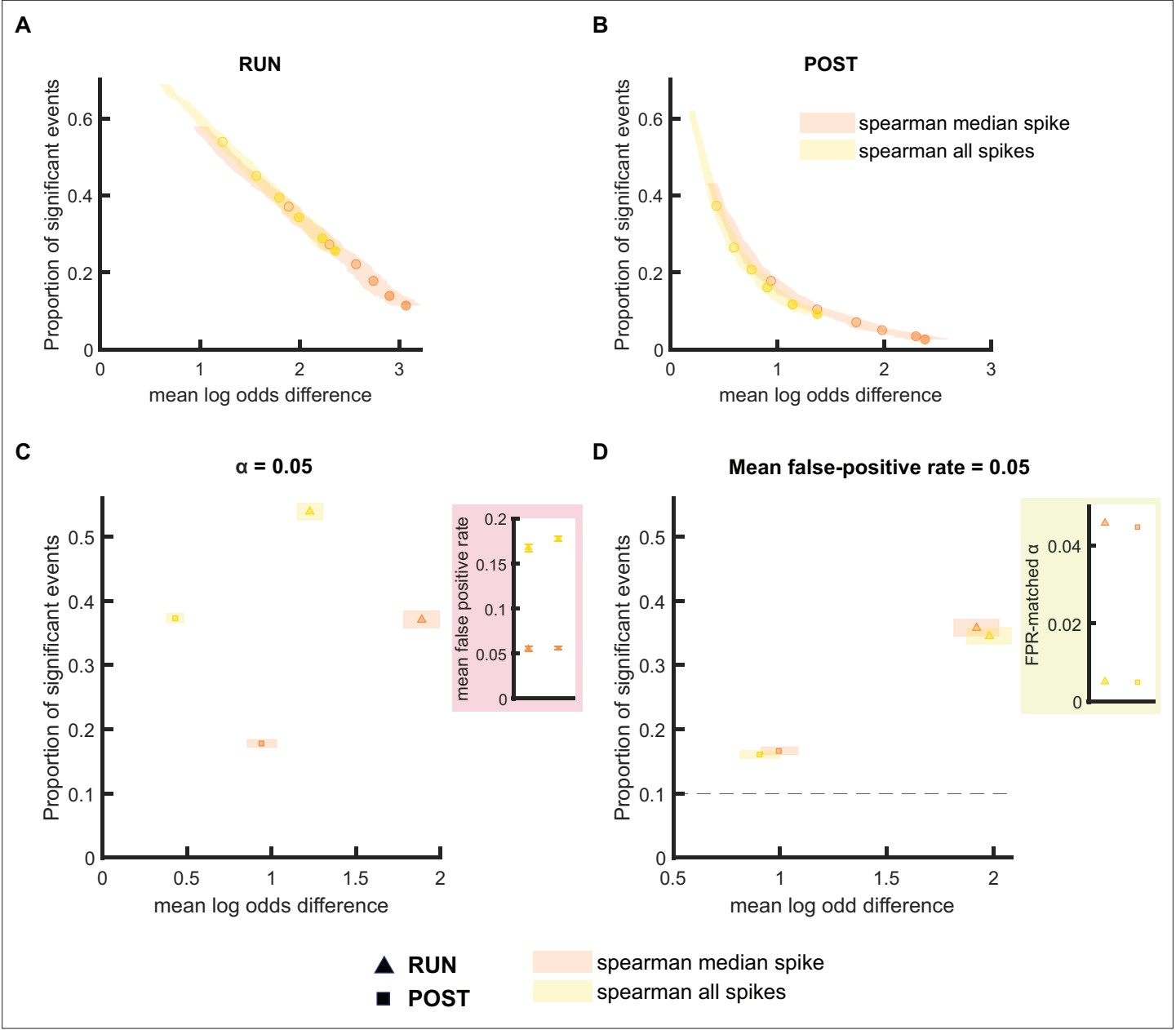

**Figure 5.** The performance of rank-order-based replay detection method depends on the selection of spikes for analysis. (**A,B**) The proportion of significant events and mean log odds difference at different alpha levels (0.2–0.001) when (1) all spikes or (2) only the median spike fired by each place cell was included for rank-order-based replay analysis. The shaded region indicated 95% bootstrap confidence interval for mean log odds difference. The six dots with increasing color intensity for each distribution represent the data at an alpha level of 0.05, 0.02, 0.01, 0.005, 0.002, and 0.001. (**A**) Replay events detected during RUN. (**B**) Replay events detected during POST. (**C,D**) The proportion of significant events and mean log odds difference at (**C**) an alpha level = 0.05 and (**D**) an FPR-matched alpha level with a mean false-positive rate of 5%. The shaded box indicates a 95% bootstrap confidence interval for both the proportion of significant events and mean log odds difference. The triangle symbol is used to represent replay events during RUN and the square symbol was is to represent replay events during POST. The dashed line represents the approximate chance level at mean false-positive rate of 5%. (Number of candidate replay events: RUN n = 4643 and POST n = 15283). The 95% confidence interval for the proportion of significant events, mean log odds difference, mean false-positive rates, and the FPR-matched alpha level for rank-order-based methods using all spikes or only median spike within the replay event are available in *Figure 5—source data 1*.

The online version of this article includes the following source data and figure supplement(s) for figure 5:

**Source data 1.** Summary of replay detection performance for rank-order-based methods.

**Figure supplement 1.** The mean false-positive rate across both tracks when all spikes or median spike fired by each place cell was included for Spearman's rank-order-based analysis.

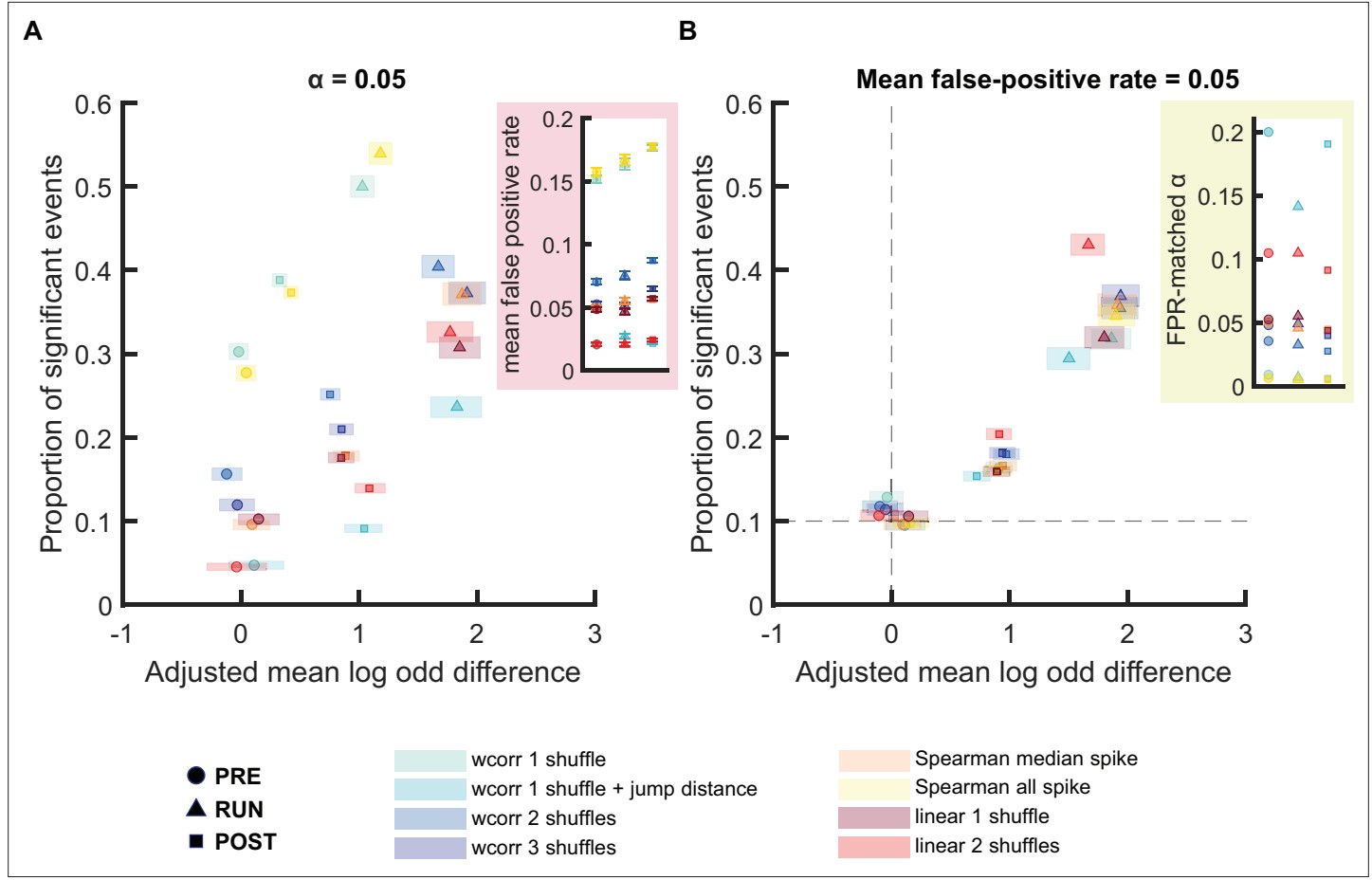

**Figure 6.** Comparison of different replay detection methods for replay events during PRE, RUN, and POST. (**A,B**) The proportion of significant events and mean log odds difference at (**A**) an alpha level = 0.05 and (**B**) an FPR-matched alpha level with a mean false-positive rate of 5% using a range of different methods: (1) weighted correlation with place bin circular shuffle, (2) weighted correlation with place bin circular shuffle and jump distance threshold at normalized track length 0.4, (3) weighted correlation with place field circular shuffle and time bin permutation shuffle, (4) weighted correlation with place field circular shuffle, spike train circular shuffle, and time bin permutation, (5) Spearman's rank-order based correlation using only median spike fired by each place cell, (6) Spearman's rank-order based correlation using all spikes fired by each place cell, (7) linear fitting with place bin circular shuffle, (8) linear fitting with place field circular shuffle and time bin permutation shuffle. The shaded box indicated 95% bootstrap confidence interval. The circle symbol was used to represent replay events during PRE. The triangle symbol was used to represent replay events during RUN and the square symbol was used to represent replay events during POST. Inset plots are (**A**) the mean false-positive rate across shuffling methods and (**B**) the FPR-matched alpha level across shuffling methods. The dashed line in (**B**) represented the approximate chance level detection rate and track discriminability at mean false-positive rate of 5%. (Number of candidate replay events: PRE n = 8485, RUN n = 4643 and POST n = 15283). The 95% confidence interval for the proportion of significant events, mean shuffled-corrected log odds difference, mean false-positive rates, and the FPR-matched p-value across all methods are available in *Figure 6—source data 1*.

The online version of this article includes the following source data and figure supplement(s) for figure 6:

**Source data 1.** Summary of replay detection performance across multiple detection methods.

**Figure supplement 1.** Comparison of different replay detection methods for replay events during PRE, RUN, and POST when log odds difference was not shuffle-subtracted.

**Figure supplement 2.** Comparison of mean log odds difference using weighted correlation and linear fitting when the posterior probabilities were normalized within track or cross-track.

**Figure supplement 3.** Comparison of replay detection performance using weighted correlation for replay events during PRE, RUN, and POST.

**Figure supplement 4.** Comparison of replay detection performance using Spearman's rank-order-based correlation for replay events during PRE, RUN, and POST.

**Figure supplement 5.** Comparison of replay detection performance using linear fitting for replay events during PRE, RUN, and POST.

**Figure supplement 6.** Comparison of the proportion of multi-track events and mean false-positive rate for different replay detection methods, and for replay events during PRE, RUN, and POST.

# Discussion

Previous studies describing the phenomenon of replay have used a wide variety of methods and criteria for replay detection. The most common methods in the last decade have been a rank-order correlation (applied to spike trains), and the scoring of decoded trajectories (using a weighted correlation or linear fit) (*Foster, 2017*; *Tingley and Peyrache, 2020*; *van der Meer et al., 2020*). However, even within these two classes of methods, there is substantial variation in the parameters and strategies for statistical testing, including the type and number of shuffling procedures, and the use of ripple power or jump distance as a threshold. The diversity in replay methods prevents an unbiased comparison between different replay findings. In particular, while replay detection statistical tests are commonly used with an alpha level of 0.05, this value does not necessarily match the actual false-positive detection rate for all methods. This issue is not just limited to when the entire spike train is used to calculate the rank-order correlation (as discussed above), but also applies to all methods using decoded neural activity. The use of either spatially smoothed place fields or spike bursts spanning multiple time bins similarly violates the assumption of independent samples. While the shuffling of neural patterns comprising the replayed trajectory may compensate for non-independent samples, its effectiveness varies substantially, especially given the wide range of parameters used across the published replay literature. Furthermore, while imposing stricter criteria for replay detection would theoretically improve the average fidelity of the events and reduce the false-positive rate, it comes with the caveat of potentially increasing the rejection of true replay events.

Quantifying and comparing replay detection performance has been an ongoing challenge due to the lack of a ground truth. Taking advantage of data collected from rats running on two linear tracks, here we introduce a novel framework for comparing different sequence-based replay detection methods - quantifying how well a given replay detection method correctly discriminates track-specific replay events. In particular, this framework quantifies the proportion of significant replay events (quantity) and the mean log odds difference (track discriminability) of a collection of track-specific replay events detected as well as adjusting the alpha level used for replay detection according to an estimated false-positive rate of 5% based on cell-id randomized dataset. Using this approach, we make six key observations: (1) For candidate events selected using an MUA threshold, the additional step of including a ripple power threshold is not necessary for RUN replay but is recommended for POST replay events. For our dataset, the POST replay events with ripple power below a z-score of 3–5 were indistinguishable from spurious events. While the exact ripple z-score threshold to implement may differ depending on the experimental condition (e.g. electrode placement, behavioral paradigm, noise level, etc.) and experimental aim, our findings highlight the benefit of using a ripple power threshold for detecting replay during POST. (2) More shuffles, with a preference for pre-decoding shuffles, lead to better replay detection, even after the alpha level is adjusted to match an estimated false-positive rate. (3) Without adjusting the alpha level, there is a very wide distribution of both the track discriminability and the proportion of significant events detected across methods. False-positive rates also vary substantially, and one should not assume that replay exists simply because the proportion of significant events detected is greater than the alpha level (typically 5%) used in detection. (4) After adjusting the alpha level to match an estimated false-positive rate, there is greater similarity observed between methods, however using either a weighted correlation or a linear fit replay scoring with only one shuffling procedure still results in the worst overall performance for our dataset. (5) Our metric for replay track discriminability yields a similar result across many methods once the alpha level is adjusted to fix the estimated false-positive rate at 5%, however the proportion of replay events detected remains more variable. (6) Regardless of the method used after the alpha level is adjusted, preplay events are at chance level for both track discriminability (mean log odds difference of 0) and proportion detected (10% of events across two tracks). Previous studies have used an alpha level of 0.05, which in our hands substantially changes the proportion of replay events detected across methods - overly strict methods such as weighted correlation with a jump distance criterion would have a proportion less than 5%, while more lenient methods such as rank-order correlation with all spikes included can have a proportion closer to 30%. Therefore, without actually measuring a false-positive rate, one cannot assume that a higher proportion of events relative to the alpha level is evidence, on its own, for the existence of replay. In all methods tested, we do not see any evidence of preplay events behaving differently from spurious replay events generated using randomized data. In contrast to this, both POST and RUN replay events have

a proportion of significant events substantially above the chance level for detection, for any method used with an FPR-matched alpha level.

Here, we have focused on a number of common replay scoring methods and criteria, however there are many variations and nuanced differences between published methods, that were beyond the scope of this study to test. For instance, we did not examine the impact of spatial smoothing (place fields) and temporal smoothing (posterior probability matrix and/or spike train) on replay detection. For this study, we avoided any Gaussian smoothing of the posterior probability matrix in our analysis but opted to use relatively large bin sizes for Bayesian decoding (10 cm by 20 ms). It is possible that with smaller bin sizes, replay detection could improve, or yield a different outcome when comparing methodologies. However, as the size of the decoding bin decreases, noise in the decoded trajectory will likely increase or alternatively must be removed using another smoothing method, which in turn may change the false-positive rate of detection. Replay experiments can often vary along many other parameters (e.g. number of neurons, distribution of place fields, length of track, etc.), and it is possible that these differences could also affect the false-positive rate. As such, it is critical for replay studies to independently verify and report an estimated false-positive rate even for the experiments involving only one spatial context.

One caveat associated with this framework is that use of any kind of randomized dataset may potentially underestimate the false-positive rate of replay detection in a way that is biased toward specific shuffling distributions (i.e. place-based or time-based shuffling). In case of a cell-id randomization, each cell's firing statistics of replay events is preserved, but the relationship between each cell's spike train and its place fields is randomly swapped, which may be potentially biased more toward place-based shuffling. Generally, we observed that the cell-id randomized dataset did not underestimate the empirically estimated false-positive rates compared to place field shifted dataset and spike train shifted dataset and performed similarly to the cross-experiment shuffled dataset (*Figure 1—figure supplement 4*). Even for the case of place field shuffling where the spike train shifted dataset resulted in a higher false-positive rates when compared to the cell-id randomized dataset, the difference between cell-id randomized FPR and spike train randomized FPR is much smaller than that between the place field randomized and spike train randomized FPR at an alpha level of 0.05. This suggests that while cell-id randomization is relatively more sensitive to place-based shuffling compared to time-based shuffling, such bias was relatively minor. Taken together, our results suggest that, while imperfect, cell-id randomization (or alternatively a randomization using cross-experiment shuffled place fields) is the relatively more conservative and universally applicable null distribution for empirically estimating false-positive rates given the detection methods used here. Nevertheless, our finding about the performance of different replay detection methods should be interpreted with care. For example, while alpha level adjustment-based empirically estimated FPR allowed us to compare different detection methods, the comparison was made between distributions of candidate replay events rather than based on each individual event. Therefore, two methods with similar performance in terms of track discriminability and proportion of significant events do not necessarily detect the same sets of events as significant. It is also important to note that alpha-level adjustment implemented in this framework is not meant to be a way to optimize detection because it does not directly address method-specific issues and remove true false-positives caused by effects such as edge effects and neuronal spiking statistics.

The use of two linear tracks was critical for our current framework to calculate track discriminability. While this would limit the generality of our approach as most replay studies involved only one context rather than multiple contexts, with modification it is possible that this framework might be adapted to single-track data. For example, track discriminability could theoretically be calculated using a cell-id permutation to create a virtual second track or by comparing the discriminability across different portions of the track (depending on the experimental design), although exploring these options were beyond the scope of our current study.

One key consideration is that due to algorithmic constraints, some sequence detection methods may be biased to detecting certain types of replay trajectories. For instance, a linear fitting method may be more prone to detect linear replay trajectory compared to a sigmoidal-shaped or jumpy replay trajectory. On the other hand, a rank-order correlation analysis of spike trains is more relaxed in terms of the rigidity of the temporal structure, and therefore is more likely to include non-linear events with gaps. As such, there may be other factors to consider beyond track discriminability and the proportion

of detected events when choosing a replay detection method. However, it is interesting to note that despite substantial differences in these algorithms used for replay detection, all methods had a similar track discriminability after adjusting the alpha level to a fixed estimated false-positive rate of 5%.

Last but not least, in addition to comparing the performance of different replay quantification strategies, this framework can also be applied to compare replay events detected across different behavioral states or experimental conditions. Previously, differences in replay across conditions have been measured using the rate or number of replay events (*Gillespie et al., 2021*; *Huelin Gorriz et al., 2023*; *Ólafsdóttir et al., 2015*; *Xu et al., 2019*), or decoding track bias (*Carey et al., 2019*; *Tirole et al., 2022*). However, combining these two approaches creates a more comprehensive metric for understanding the difference in replay across different conditions.

## Methods

All the behavioral and electrophysiological data used in this study were previously described in detail in *Tirole et al., 2022*.

### Behavioral paradigm and electrophysiological recording

Five male Lister-hooded rats (350–450 g) were implanted with 24 independently movable tetrodes, using a custom-made microdrive. In three rats, the microdrive was split into two 12 tetrodes groups, aimed at the right and left dorsal CA1 regions of the hippocampus respectively (AP: –3.48 mm; ML: ±2.4 mm from Bregma). In the remaining two rats, right dorsal CA1 was targeted with 16 tetrodes (AP: 3.72 mm, ML: 2.5 mm from Bregma), with the remaining 8 tetrodes targeting visual cortex.

Before the experimental sessions, each animal was trained for approximately 2 days (30 min sessions) to run back and forth on a linear track. The animal was motivated to run by receiving a reward (chocolate-flavored soy milk) at each end of the track. The training took place in a different room and the track configuration used for training was distinct from the experimental linear tracks.

Each recording session started with a 1 hr rest (PRE) epoch in which the rats were placed in a previously habituated rest pot at a remote location. The rest pot was made of circular enclosure (20 cm diameter) surrounded by a 50 cm tall black plastic wall, which prevented the rats from viewing rest of the environment. Following the PRE epoch, the rats went through one of the two protocols described below, for which both were followed by a POST rest epoch in the rest pot:

1. Rats were exposed to two novel linear tracks (2 m long).
2. Rats were exposed to three novel tracks (2 m long). Data from the first track is not included here to create a two-track experiment consistent with the remaining data.

### Place cell analysis

Place cells were required to have a minimum peak firing rate greater than 1 Hz, based on the unsmoothed ratemap. Spike trains used to compute the place field were speed filtered (4–50 cm/s).

### Bayesian decoding of spatial location

A naïve Bayesian decoder was used to estimate the brain's estimation of animal's position during behavior and virtual position during a replay event, using non-overlapping 20 ms time bins, and 10 cm position bins.

$$P\left(x|n\right) = CP\left(x\right)\left(\prod_{i=1}^{N} f_i\left(x\right)^{n_i}\right) exp\left(-\tau \sum_{i=1}^{N} f_i\left(x\right)\right)$$

where $P(x|n)$ is the probability of the animal being at a specific position given the observed spiking activity, $C$ is a normalization constant, $x$ is the animal's position, $f_i(x)$ is the firing rate of the $i$th place field at a given location $x$, and $n$ is the number of spikes in the time window $\tau$. The normalization constant was defined as the summed posterior probabilities across both tracks.

### Detection of candidate replay event

Candidate replay events were first selected based on MUA, which was smoothed with a Gaussian kernel (sigma = 5 ms) and binned into 1 ms steps. Only events with MUA bursts with a maximum

duration of 300 ms and z-scored activity over 3 were selected. Furthermore, ripple power was used as an additional criterion for candidate event selection. Ripple-band (125–300 Hz) LFP signal was smoothed with a 0.1 s moving average filter. Unless it was mentioned otherwise, the candidate events were discarded if the peak ripple power was less than z-score of 3. Moreover, the candidate events were further thresholded such that only events where the animal's running speed was less than 5 cm/s, at least five active place cells and event duration over 100 ms and below 750 ms, were included for subsequent analysis. To maximize detection of replay events and avoid discarding minority events due to noisy probability decoding at the beginning or the end of the event, candidate replay events were split into two segments where the midpoint was determined based on the minimum MUA activity in the middle third of the candidate event. Both segments were decoded and analyzed for statistical significance independently with the alpha level adjusted to half of the alpha level used for 'full' event (e.g. $\alpha = 0.05 \rightarrow \alpha = 0.025$). It should be noted that same criteria for candidate event inclusion such as minimum event duration and active place cell number still applied to these 'half' events. For this study, RUN replay was also defined as awake replay events when the animals were physically on the linear tracks (with moving speed less than 5 cm/s). PRE and POST replay was defined as replay events that took place during rest and sleep periods (not differentiated from each other) within the rest pot before and after experiencing both novel tracks, respectively.

## Sequence-based replay scoring

For this study, two different sequence quantification methods were used to quantify the sequenceness of the decoded posterior probability matrix for each event.

### Weighted correlation

Weighted correlation method calculated the correlation coefficient between the change in decoded posterior probability bins across position and time by weighing each estimated position by its decoded posterior probability:

Weighted mean:

$$m\left(x; prob\right) = \frac{\sum_{i=1}^{M} \sum_{j=1}^{N} prob_{ij} x_i}{\sum_{i=1}^{M} \sum_{j=1}^{N} prob_{ij}}$$

Weighted covariance:

$$cov\left(x, t; prob\right) = \frac{\sum_{i=1}^{M} \sum_{j=1}^{N} prob_{ij}\left(x_i - m\left(x; prob\right)\right)\left(t_j - m\left(y; prob\right)\right)}{\sum_{i=1}^{M} \sum_{j=1}^{N} prob_{ij}}$$

Weighted correlation:

$$corr\left(x, t; prob\right) = \frac{cov\left(x, t; prob\right)}{\sqrt{cov\left(x, x; prob\right) cov\left(t, t; prob\right)}}$$

where $x_i$ is the $i$th position bin, $t_j$ is the $j$th time bin, and $prob_{ij}$ is the probability at the position bin $i$ and time bin $j$.

### Linear fitting

The linear fitting method finds the line of best fit that describes the decoded linear trajectory for each candidate event. In practice, this method would calculate all the possible linear kernel at different slopes (from 100 to 5000 cm/s) and intercept and then sum all the probabilities within 10 cm below or above each fitted line as a goodness of fit score:

$$R\left(slope, intercept\right) = \frac{1}{n} \sum_{t=0}^{n-1} prob\left(\left|pos\left(t\right) - \left(slope \times t \times T + intercept\right)\right| < d\right)$$

where $t$ is each time bin, $prob$ is the decoded posterior, $pos$ is the position bin, $T$ is the time bin, $d$ the maximum distance from the line of fit.

The combination of slope and intercept that maximizes the posterior decoded probabilities along the potential linear trajectory would be considered as the line of best fit.

A similar sequence-based analysis was applied to the randomized dataset (obtained by a cell-id permutation where the cell identity for each spike train in a *candidate* replay event was randomized before decoding). This shuffle was designed to preserve the firing statistics across place cells while randomizing the order of place fields along the track, in turn disrupting any relationship with the spike sequences produced during candidate events. For this study, to enhance the accuracy of our false-positive rate estimation, given that false-positive events were generally less frequent than replay events detected during POST and RUN in our original dataset, we generated three cell-id randomized candidate events for each real candidate event (with a new set of random seeds for every event) such that the total number of cell-id randomized events would equal three times the total number of real candidate events.

Following obtaining a sequence score (either a weighted correlation or linear fit-based score) for each event, we then calculated the p-value by comparing the event sequence score relative to the four shuffled distributions, each designed to randomize certain aspects of the sequential place cell firing.

## Pre-decoding shuffling procedures

Shuffle 1: Spike train circular shuffle, in which the spike count vectors for each cell were independently circularly shifted in time by a random amount within each replay event, prior to decoding. This shuffle was designed to degrade the temporal order of the neuronal spiking with minimal disruption of each neuron's spiking statistics and spatial template.

Shuffle 2: Place field circular shuffle, in which each ratemap was circularly shifted in space by a random amount of position bins prior to decoding. This shuffle was designed to randomize the preferred firing locations while preserving the temporal structure of the spiking patterns.

## Post-decoding shuffling procedures

Shuffle 3: Place bin circular shuffle, in which posterior probability distribution for each time bin (column) was independently circularly shifted by a random amount. This shuffle was designed to shift the decoded position at each time bin while preserving each neuron's spiking properties.

Shuffle 4: Time bin permutation shuffle, in which the order of the time bins within each event was permutated randomly. This shuffle was designed to disrupt within-event temporal structure without interfering each neuron's place field template.

Each type of shuffle was performed 1000 times for each event to obtain a shuffle distribution of sequence scores.

## Rank-order correlation

In addition to a probabilistic decoding-based approach, a rank-order correlation approach (Spearman's correlation coefficient) was also performed directly to the spike train data during candidate replay events using all spikes (method 1) or only the median spike time (method 2) produced by each place cell:

$$\rho \left(place\ cell\ order, spike\ time\right) = 1 - \frac{6 \sum d^2}{n\left(n^2 - 1\right)}$$

where *d* is the difference between the ranks of order of place cells that are active during a given event and the observed spike time, *n* is the number of spikes included for analysis (i.e. all spikes or only median spike produced by each place cell), and $\rho$ is the Spearman's rank correlation coefficient.

The p-value for Spearman's $\rho$ was computed by comparing the original $\rho$ relative to the shuffled distribution where the order of spike was randomly permuted.

## Quantification of significant event proportion and false-positive rate

While standard replay analysis used $p < 0.05$ (i.e. score greater than 95% of the shuffled distribution) as the cut-off point for statistical significance, we calculated the proportion of significant events using alpha level ranging from 0.2 to 0.001. When more than one shuffle type was used together for detection, the sequence score was required to be below the alpha level for each shuffle type used, in order for the event to be considered significant. In cases when more than one shuffle was used to calculate

the p-value, the highest p-value would be used to represent the sequenceness of each significant replay event. In cases where the candidate replay event was detected as statistically significant for both tracks (referred to as 'multi-track event'), the candidate event was assigned as both track 1 and track 2 events. However, each multi-track event would be only counted once when calculating the proportion of significant events to avoid double counting. Assuming false-positive rate at 5%, we would expect approximately 10% detection rate (significant events labeled as either track 1 or track 2) with the caveat that this value can potentially decrease in the case where a high number of multi-track events were detected.

$$Proportion\ of\ significant\ event = \frac{\left(n\left(significant\ Track\,1\ event\right) + n\left(significant\ Track\,2\ event\right) - n\left(multitrack\right)\right)}{n\left(all\ canditate\ events\right)}$$

For cell-id randomized replay events, similar analysis was performed to calculate mean false-positive rate across both tracks.

The mean estimated false-positive rates across both tracks (later used for alpha level adjustment) were calculated by adding the number of false-positive events on both tracks divided by 2 (i.e. number of tracks), which was then divided by the total number of candidate events:

$$mean\ false\ positive\ rate = \frac{n\left(false\ positive\ Track\,1\ event\right) + n\left(false\ positive\ Track\,2\ event\right)}{2 \times n\left(all\ canditate\ events\right)}$$

## Sequenceless decoding and quantification of track discriminability

In order to cross-check the quality of replay events detected based on sequence-based approach, we used an independent metric that measure reactivation bias based on sequenceless decoding (*Carey et al., 2019*; *Tirole et al., 2022*). Only cells with stable place fields in the first and second half of the behavioral episode on both tracks (peak in-field firing rate >1 Hz) were included in our sequenceless decoding analysis. Prior to Bayesian decoding, ratemaps for track 1 and track 2 were concatenated as a single matrix $\left[nCell \times \left(nPos\left(Track\,1\right) + nPos\left(Track\,2\right)\right)\right]$, where *nPos* is the number of position bin on each track (i.e. 20×10 cm$^2$ position bins for both tracks). For a given replay event, the decoded posterior probabilities for each time bin were normalized across all position bins from both tracks to sum to 1. We quantified the logarithmic ratio between the summed posterior probabilities of both tracks, and then z-scored this ratio relative to a shuffled distribution computed by Track ID shuffle.

$$log\ odds = log\frac{\sum_{i=1}^{M}\sum_{j=1}^{N}prob_{ij}\left(Track\,1\right)}{\sum prob\left(Track\,2\right)}$$

$$m\left(x;prob\right) = \frac{\sum_{i=1}^{M}\sum_{j=1}^{N}prob_{ij}\left(Track\,1\right)}{\sum_{i=1}^{M}\sum_{j=1}^{N}prob_{ij}}$$

For each place cell in the track ID shuffle, its place field templates for track 1 and track 2 were randomly assigned (i.e. each cell's ratemap was either swapped or not swapped). Then, we quantify the track discriminability in terms of the difference in mean log odds between track 1 and track 2 replay events, originally detected using sequence-based replay detection methods, to cross-check the quality of replay content.

## Alpha level adjustment based on mean false-positive rate across both tracks

After calculating the mean false-positive rate across both tracks at alpha levels ranging from 0.2 to 0.001, we then identify the alpha level that leads to the mean false-positive rate closest to 0.05 for a given method, which would be referred to as FPR-matched alpha level. The log odds difference and proportion of significant events detected at the FPR-matched alpha level would be used for the subsequent method comparisons.

## Statistics

### Bootstrapping for method comparisons

To determine if the differences between two methods (i.e. mean log odds or proportion of significant events) were statistically significant, we used a bootstrapping procedure where candidate replay events were resampled with replacement 1000 times. This created bootstrapped distributions of log odds differences, the proportion of significant events detected, and false-positive rates at each alpha level ranging from 0.2 to 0.001.

When comparing the proportion of significant events and log odds difference at the original alpha level = 0.05, we calculated the 95% confidence interval for both metrics obtained at an alpha level = 0.05. When comparing the proportion of significant events and log odds difference at the FPR-matched alpha level, we calculated the 95% confidence interval for both metrics obtained at the alpha level when the mean false-positive rate was closest to 5%. The difference between the two boot-strapped distributions was only considered statistically significant when the 95% confidence intervals did not overlap.

### Bootstrapping for log odds significance

To determine if the log odds difference of the replay events detected during PRE, RUN, and POST by a given method was statistically significant from chance level track discriminability, we calculated the confidence interval for the difference between bootstrapped distribution of the original replay data and the cell-id randomized replay data. The mean difference between two bootstrapped distributions was only considered statistically significant when the 95% confidence interval did not overlap with 0.

## Acknowledgements

We thank members of the Bendor Lab and Henrik Singmann for valuable discussion and Tom Wills and Aman Saleem for their comments on the manuscript. Rat schematic in *Figure 1A* was adapted with permission from SciDraw.io (https://doi.org/10.5281/zenodo.3926277, https://doi.org/10.5281/zenodo.3926237, https://creativecommons.org/licenses/by/4.0). This work was supported by the Medical Research Council scholarship (MR/N013867/1) (MTak), the European Research Council starter grant (CHIME) (DB), the Human Frontiers Science Program Young Investigator Award (RGY0067/2016) (DB), and the Biotechnology and Biological Sciences Research Council Research grant (BB/T005475/1) (DB). The Titan Xp used for this research was donated by the NVIDIA Corporation.

## Additional information

### Funding

| Funder | Grant reference number | Author |
|---|---|---|
| Medical Research Council | Graduate student scholarship MR/N013867/1 | Masahiro Takigawa |
| European Research Council | Starter Grant CHIME | Daniel Bendor |
| Human Frontier Science Program | Young Investigator Award RGY0067/2016 | Daniel Bendor |
| Biotechnology and Biological Sciences Research Council | Research Grant BB/T005475/1 | Daniel Bendor |

The funders had no role in study design, data collection and interpretation, or the decision to submit the work for publication.

### Author contributions

Masahiro Takigawa, Conceptualization, Software, Formal analysis, Funding acquisition, Visualization, Methodology, Writing - original draft, Writing - review and editing; Marta Huelin Gorriz, Margot Tirole, Software, Investigation, Methodology, Writing - review and editing; Daniel Bendor, Conceptualization,

Software, Supervision, Funding acquisition, Methodology, Writing - original draft, Writing - review and editing

## Author ORCIDs
Masahiro Takigawa ⓘ https://orcid.org/0000-0002-0162-9017
Marta Huelin Gorriz ⓘ https://orcid.org/0000-0002-0281-0627
Margot Tirole ⓘ https://orcid.org/0000-0003-0674-6690
Daniel Bendor ⓘ https://orcid.org/0000-0001-6621-793X

## Ethics
All experimental procedures and post-operative care were approved and carried out in accordance with the UK Home Office, subject to the restrictions and provisions contained within the Animal Scientific Procedures Act of 1986. Experiments were conducted under PPL P61EA6A72 (Bendor).

## Decision letter and Author response
Decision letter https://doi.org/10.7554/eLife.85635.sa1
Author response https://doi.org/10.7554/eLife.85635.sa2

---

## Additional files

### Supplementary files
• MDAR checklist

### Data availability
Analysis code is available at https://github.com/bendor-lab/replay_detection_cross_validation (copy archived at *Takigawa, 2024*).

The following previously published dataset was used:

| Author(s) | Year | Dataset title | Dataset URL | Database and Identifier |
| --- | --- | --- | --- | --- |
| Tirole M, Huelin M, Takigawa M, Kukovska L, Bendor D | 2023 | Experience-driven rate modulation is reinstated during hippocampal replay | https://doi.org/10.5061/dryad.ksn02v76h | Dryad Digital Repository, 10.5061/dryad.ksn02v76h |

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
