## [Editor Report]

In this valuable work, the authors present a case for a new standard in replay detection, tackling the formidable problem that different methods can produce in vastly different results. The authors show compelling evidence about the source of this problem (which is that the true false positive rate can vary wildly between methods). The authors present a solution to the challenge of underestimation of the false positive rate and leverage new experimental data and novel analysis techniques to provide solid evidence that – under specific assumptions – their approach is effective.

---

## [Decision Letter]

**Decision letter after peer review:**

Thank you for submitting your article "Evaluating hippocampal replay without a ground truth" for consideration by *eLife*. Your article has been reviewed by 3 peer reviewers, and the evaluation has been overseen by a Reviewing Editor and Laura Colgin as the Senior Editor.

The reviewers have discussed their reviews with one another and multiple reviewing editors and the Reviewing Editor has drafted this to help you prepare a revised submission.

Essential revisions (for the authors):

(1) The reviewers were unanimous in disagreeing with the idea that the cell-ID shuffle could be used as a method to estimate a false positive rate, and thus generally opposed the prospective premise of the work.

(1b) There was an opinion that "it might still be a useful tool to compare popular replay scoring methods, as long as (1) instances where the threshold is incorrectly referred to as an "adjusted p-value" are removed and (2) it is not recommended as a novel replay detection approach but simply a tool to compare the methods.

(2) However, the reviewers felt that there was value in comparing track discriminability vs. sequence detection, particularly for analysis and comparisons including the PRE-RUN data. Reviewers were moderately enthusiastic about an alternative presentation that concentrated on reporting effect sizes rather than p-values alone, and shifted towards understanding the relationships between the factors.

(3) Towards this end, in the discussion, the reviewers were also concerned about the idea of restricting analyses only to those neurons that had fields in both environments. This makes sense as a potential way to avoid concerns about spike sorting, but a subsequent paper should include the obviously functionally important population of neurons that have place fields in only one environment.

*Reviewer #1 (Recommendations for the authors):*

1. I would suggest showing the analysis on a single environment and three environments if possible for the log odds measure.

2. It would be ideal for the functions used for the main analysis to be clearly documented in the code repository. At the very least highlighted in the README for use by others.

3. Do these results hold over animals? It would be important to see the variability in conclusions if possible.

4. Would the conclusions about p-values hold if you applied a multiple comparisons correction like the Benjamini and Hochberg procedure or Storey's q-value first? I would suggest clarifying why the empirically determined adjusted p-value works better than the other multiple comparison measures.

5. I would clarify that the p-value threshold and significant events are determined by the replay score only in the text.

6. "computing the summed posteriors (across time bins) within the replay event for each track" – Please clarify or correct this line because I believe the posterior is summed across time and position.

7. "difference in mean log odds between track 1 and track 2 replay events" – It is unclear what this mean is over. Session? Over all animals?

8. " ripple power threshold is not necessary for RUN replay, but should be applied to POST replay events" – It is my understanding that a multiunit threshold is still applied. It would be important to clarify this if this is the case.

9. "a more positive mean log odds difference would indicate a higher trajectory discriminability in the replay content, and a higher confidence in replay quality. In contrast, a mean log odds difference of 0 would suggest that the quality of replay events is comparable to chance level trajectory discriminability, and that the detected event population most likely consists of low fidelity events, indistinguishable from false-positives,"- is there an absolute value being taken here? The measure can be both positive and negative, so unless the track selected by the replay score is always the positive going track, this cannot be quite correct. Please clarify if that is the case and note that the statistic can be negative.

10. The term "cross-validation" is being used improperly. Cross-validation has a specific meaning in statistics, implying a test and training set. I would suggest using validation, or orthogonal validation instead.

11. I would add individual examples of the log odds procedure and z-score (as in Tirole et al. 2022, Figure 4B) to help the readers understand the procedure better.

12. "Based on the desire to maximize the statistical independence of data, we avoided any smoothing in our analysis, as this could increase the false-positive rate of replay detection. Because we did not use spatial or temporal smoothing, we opted to use larger bin sizes for Bayesian decoding (10 cm by 20 ms)." – Using bins of any kind is a form of "smoothing" and choice of bin edges will lead to different answers. Furthermore smoothing is not inherently bad as it can reduce variance and potentially lead to more statistical power. Choosing not to smooth with a gaussian is fine as a choice but I would change this sentence.

13. "To determine if the decrease in detection rate was associated with decrease in the false-positive rate for this detection method, we calculated the false-positive rate (mean fraction of spurious replay events detected across both tracks)" – I would suggest clarifying that the spurious replay events detected are from the shuffled data. It is clear in the figure legend, but not in the main body of the text.

*Reviewer #2 (Recommendations for the authors):*

I would recommend the authors focus on the analysis of preplay in their dataset. They can better analyze their observation that no reactivation is evident in events with high replay scores, which will be of interest to the field. I believe that the rest of the work is highly problematic.

*Reviewer #3 (Recommendations for the authors):*

I have two fundamental suggestions.

1) While I agree that the sequenceless decoding is important to cross-validate the main findings of the paper, reading the paper for the first time gave me the (false) impression that the proposed framework of detecting replay somehow uses this measure to select the final replay events. After reading the methods multiple times, it seems to me that first impression was untrue, and the proposed replay detection only depends on adjusting the p-value based on the empirical false positive estimate using a cell-ID shuffle. If this is correct, then the sequenceless decoding only provides an x-axis as additional evidence to help convince the reader of the conclusions of the comparisons.

The way the paper is written, I think most readers would make the same initial assumption as I did. The first time false positive estimates are mentioned in the results of the paper, we are directed to Figure 1D which describes the sequenceless cross-validation. Moreover, given the emphasis of this sequenceless approach in the abstract and the entire manuscript, it is easy to conclude that the false-positive rate is estimated as the number of events that don't show sequenceless log-odds higher than the shuffle.

This is problematic because many people in the field might dismiss the findings by thinking "Well that's nice but I don't have two tracks, so I can't do any of that". The only place in the manuscript that made me dig deeper was in the discussion, where the authors state "As such, it is critical for replay studies to independently verify and report this false-positive rate even for the experiments involving only one spatial context." Only after reaching this sentence did I even consider that estimating the false positive rate might not require two tracks, and it took quite some time to make sure it doesn't, and yet even now a part of me is still unsure that I got it right. I would urge the authors to make it more clear that the false positive rate can be estimated readily using a simple shuffle and does not require having two tracks.

It is my opinion that estimating the false positive rate and taking it into account is very powerful and can be a game changer for replay detection, but it won't happen if people don't understand it was done and how it was done.

2) In my initial reading of the paper, I was confused by Figure 4D (and all the figures like it). It sounds counter-intuitive that a more strict method (e.g. 2 shuffles vs 1 shuffle) can end up detecting *more* replay events. It is not trivial point to grasp, and it is easy for readers to get lost here (e.g. the y-axis "proportion of significant events" may be considered by some confused readers to mean some measure of specificity like "among the events selected by each method, how many of them were true positives"). I think something can be added to this section to better guide readers and explain what is being proposed.

I'm thinking of something like "Before controlling for the false positive rates associated with each method, as expected, stricter methods found fewer replay events than more permissive methods: for example, a single shuffle selected 50% of awake events as significant, as opposed to 40% that the stricter 2-shuffles method selected (Figure 4C). However, after adjusting the p-value to control for the false positive rate, we found that stricter methods actually resulted in detecting more replay events: the single shuffle awake replay adjusted p-value was 0.007 which brought down the proportion of significant events to 32%, while the stricter 2-shuffles awake replay adjusted p-value was 0.032, resulting in 35% of events detected as significant. This shows that controlling for the empirically measured false positive rate can, almost paradoxically, result in stricter methods detecting more replay events." The "Place+jump distance" is somewhat separate and can be presented after taking the time to explain this very critical point.

[Editors' note: further revisions were suggested prior to acceptance, as described below.]

Thank you for resubmitting your work entitled "Evaluating hippocampal replay without a ground truth" for further consideration by *eLife*. Your revised article has been evaluated by Laura Colgin (Senior Editor) and a Reviewing Editor.

The manuscript has been improved but there are some remaining issues that need to be addressed, as outlined below:

The reviewers reached a consensus that the paper requires minor revisions. Please see individual reviewer comments below. In particular, there was a consensus that the discussion of the paper needs to clearly articulate a proper interpretation of the corrected p-values. Specifically, that using one shuffle (cell-ID) to correct the p-values of another (i.e., circular) on a collection of events is not the same as somehow correcting the replay detection for individual events.

*Reviewer #1 (Recommendations for the authors):*

Thank you to the authors for their thoughtful responses to the reviews. I do think the response clarified several aspects of the manuscript and improved the clarity of the manuscript overall. The documentation and improvement of the code associated with the manuscript is also improved.

As I understand the logic, the main claims of the framework to study sequence detection methods are:

1. A good sequence detection method should assign the same track as track decoding.

2. The significant replay events for a given method can be adjusted by matching the false positive rate resulting in greater track discriminability (and more conservatively identified replay events)

3. This can be used to compare replay detection methods or quality of replay event

The strengths of the work are:

1. it is more conservative in judging which sequence is a replay or "spurious" given that the two track model is true

2. it makes some interesting conclusions about the use of the ripple power and multiunit as criterion for detecting replay events.

The weaknesses of the work are:

1. The framework relies on an adjustment for track discriminability but, as the authors acknowledge, there are many ways in which the model can be misspecified. This only tests for a very specific way in which they are misspecified. And it doesn't seem to give one much insight into why a particular method is better. For example, the authors' response seems concerned about the lack of accounting for bursting in replay detection, but track discriminability does not circumvent this. The way to account for lack of independence in time bins is to explicitly account for the bursting, such as fitting the place field estimates with a self-history term. The track discriminability measure simply marginalizes over the time and position and already incorporates the misspecified encoding model.

2. This framework relies on classifying aspects of the replay event as significant or not significant at a given significance threshold. This is reliant on the particular null distribution specified.

3. The generality of the work is limited by their two track framing.

The work claims that their framework is a "unifying approach to quantify and compare the replay detection performance". The work falls short of doing this in general but I do think it accomplishes this with a much narrower scope, namely in adjusting methods that fail to discriminate between tracks appropriately in the case of two tracks.

– Are the significant events being corrected for multiple comparisons? Could the authors explain why no corrections are needed?

– I do not think the authors sufficiently addressed the question of what happens if the track discrimination model is misspecified in their response. For example, if there are truly three tracks but you only account for two?

*Reviewer #2 (Recommendations for the authors):*

While I appreciate the conceptual novelty of using the log odds ratio as a metric for testing the noisiness of replay events and comparing this metric with results from sequence analyses, unfortunately, I was not persuaded by the arguments presented in the authors' rebuttal. I think the study would benefit greatly if the authors consulted or collaborated closely with an expert on statistical methods, particularly on the application of resampling methods, to enhance the rigor of their approach. Without this, I remain concerned that this work can cause further confusion (and noise) rather than clarity in these analyses.

I believe I noted my concerns to great length in the initial review, but perhaps the authors will wish to look through the section "The problem with replay decoding: shuffles" section in Foster 2017 and note that each shuffle is permissive for some events and not others. E.g. if the circular shuffle is likely to pass through events that arise from edge effects, their study does not explain how a shift of the α level is going to fix that issue and remove false positives. Each type of shuffle will pass through different subsets of events, some of which may be "false positives," but they will be different events in each case. Simply adjusting the α levels or p-values, the solution proposed by this study, will not fix the underlying reasons for the false positives, and lead to potentially mistaken confidence in the results.

*Reviewer #3 (Recommendations for the authors):*

I find that the revised manuscript has considerably improved. I have one outstanding issue with the randomization shuffle, which can actually be alleviated if the authors update Figure D to include the 3 remaining shuffles, and a minor comment.

Outstanding question regarding the randomization shuffle:

My concerns about the cell-id shuffle were that (1) any shuffle, including this one, may actually underestimate the true false positive rate, and (2) this tendency to underestimate the true false positive rate will be exacerbated when the randomization shuffle is similar to the shuffle used for detecting replay events, resulting in a biased comparison between methods.

I think (1) is unavoidable as it would be hard to disprove any structure that could be inadvertently erased by any given shuffle. The true false positive rate may always be higher than our best estimates captured by our surrogate datasets, and our estimates are simply a lower bound of the true false positive rate. In particular for the cell-id shuffle (including the "cross experiment cell-id" shuffle), there may be some underlying structure in biological data which violates assumptions of independence that is not captured by place-field swapping. This would undoubtedly be the case when including a variety of cell types with generally different place field properties (e.g. along the dorsoventral axis or the deep/superficial sublayer). Even without different anatomical cell types, more excitable cells may be more likely to have multiple place fields as well as firing more spikes within candidate events – a potential structure that is not captured by the cell-id shuffle.

While concern (1) only needs to be mentioned (in the discussion or in a public review), concern (2) is particularly problematic because the authors compare the (FRP-matched) performance of different methods to conclude which are best. Figure D shows a comparison of two methods: place field shifting and place field swapping (with place fields within the same dataset, the randomization used throughout the manuscript, or from a different dataset, which is almost identical to the original randomization) before trajectory decoding. Figure D convincingly argues that relative to field swapping (cell-id shuffle), field shifting (place field circular shifted shuffle) is a much poorer method of randomization as it considerably underestimates the false positive rate for detection methods using similar shuffles. Note that the bias is slightly attenuated for the "place bin shuffle" (it is worse for the "place field shuffle"), demonstrating that there is a confounding gradient with larger similarities between the randomization and the detection shuffles resulting in larger bias.

According to the authors, Figure D shows that "cell-id based randomization is sufficiently independent from both place-based and temporally-based shuffles for replay detection". I cannot see how that can be concluded from the figure. The place field swapping methods may well be underestimating the false positive rate unevenly. Place field swapping is still a pre-decoding place-based shuffle, so the fact that its estimate agrees with the estimate of another pre-decoding place-based shuffle (place field shifting) does not demonstrate that these estimates are not a biased. I would like to stress that one of the authors' conclusions is that "more shuffles, with a preference for pre-decoding shuffles, lead to better replay detection." All of the approaches using "more shuffles, with a preference for pre-decoding shuffles" included place field shifting, which in my opinion is similar to the place field swapping of the cell-id shuffle.

I commend the authors for the approach of Figure D, which is a good way to address this key concern. I recommend this figure to be included in the manuscript. Moreover, if the authors were to include the other 3 shuffles (it currently includes "Shuffle 2" from the methods) and show that the cell-id does not underestimate the false positive rate more than those shuffles in a biased way, that would practically disprove this issue. Indeed, the concern is that the cell-id shuffle may be underestimating the false positive rate particularly for detection methods using pre-decoding and place-based shuffles, but if none of the other 3 shuffles produce higher false positive estimates for the lower row in Figure D, that would be convincing evidence that the cell-id randomization procedure is not biased towards detection methods employing pre-decoding place-based shuffles and strengthen the conclusions.

[Editors' note: further revisions were suggested prior to acceptance, as described below.]

Thank you for resubmitting your work entitled "Evaluating hippocampal replay without a ground truth" for further consideration by *eLife*. Your revised article has been evaluated by Laura Colgin (Senior Editor) and a Reviewing Editor.

The manuscript has been improved but there are some remaining issues that need to be addressed, as outlined below:

The introduction of the manuscript has already been significantly revised. However, the final revision requested by the reviewers is to spend a few sentences outlining not only sequence scoring, but also the fact that the manuscript utilises multiple types of surrogate data sets to establish null distributions. In particular, it is suggested that some assumptions about the utility of the cell-id shuffle surrogate be described, as well as the importance of track discriminability. Reviewer #2 (see below) is quite proscriptive; please consider their points and update the introduction to at least mention these issues up front.

*Reviewer #2 (Recommendations for the authors):*

The revision added some text regarding the limitations of the study. I was hoping that the revised manuscript would also be more explicit regarding the underlying assumptions. While I don't think reviewers should do the authors work for them, after such a lengthy review, perhaps this is the only way short of a rejection.

From what I can tell, these are the main assumptions. If I am wrong, perhaps the other reviewers, or the authors can correct me. Also perhaps there are others that I missed.

1) After recording from two tracks, events with low discriminability between these two tracks cannot be said to be replays, regardless of their sequence scores.

– Pooled across data, discriminability generally correlates with replay score, though this differs across datasets and timepoints of recording.

2) The cell-id shuffle provides the best null distribution to test against for replay.

– This is so because cell-id shuffled events show low discriminability even in instances where they have high sequence scores

3) The proportion of cell-id shuffled events that qualify as replays in any given period is similar to the proportion of events generated in the real data from hippocampus network that are likewise not actual replay but are labeled as significantly ordered.

– This therefore allows for a scaling correction to bring this proportion in line with the α level.

4) In the absence of performing cell-id shuffles, which is straightforward, the alternative prescription is to record from two tracks and adjust the apparent α to match those of cell-id shuffled events obtained during the recording

– This is so that the same relative number of events are labeled as replays, even though they may not be the actual replays.

In my view, these assumptions should be provided very clearly early in the manuscript, ideally in the introduction section, to guide readers towards a better understanding of the study.

---

## [Author Response]

Essential revisions (for the authors):1) The reviewers were unanimous in disagreeing with the idea that the cell-ID shuffle could be used as a method to estimate a false positive rate, and thus generally opposed the prospective premise of the work.

We fully understand the referees’ concerns with a cell-id randomization, and the caveats raised for using this approach for quantifying the false positive rate. However, we strongly feel that it is necessary to empirically estimate the false positive rate, given that the decoded data (or spike trains) are not generally composed of independent events, which may be inflating the false detection rate (and moreover by an unknown amount). We also believe that other methods suggested by reviewers may be problematic for this same reason. We will aim in this rebuttal to make a more sufficiently convincing argument that our original method, while imperfect, should remain in the manuscript, including a new validation of this method with an alternative approach (described below). We do not take this step lightly given that the reviewers were unanimous in disagreeing with our original approach. The response to the referee comments and our new analyses is described in detail below.

Reviewer #1 suggests an alternative multiple comparisons correction method, the Benjamini and Hochberg (BH) procedure, to adjust the α level rather than using an empirically-estimated false positive rate. To explore the use of this method we created a simple replay model- a true replay event was modelled as 10 neurons firing in a stereotyped order (e.g. A-B-C-D-E-F-G-H-I-J). Randomized events consisted of the same 10 neurons firing in a random order (e.g. C-F-J-A-H-D-E-B-G-I), from which we could empirically measure proportion of spurious events detected as a proxy for false positive rate. We next simulated 10000 different randomized events, varying the number of spikes produced each time the neuron fired (one spike or a burst of two or three spikes). Spikes within a burst always remained together, so for example a randomized sequence with bursts of two spikes could be- C-C-F-F-J-J-A-A-H-H-D-D-E-E-B-B-G-G-I-I. From this we quantified each event’s Spearman’s rank order correlation using all spikes (Author response image 1). We used this form of replay detection, as the statistical properties of a rank order correlation are well understood, and we can violate the assumption of independent events by replacing neurons firing single spikes with neurons firing bursts of 2 or 3 spikes. Note that this violation only occurs if we calculate the Spearman correlation coefficient with all spikes (rather than just the first or median spike for each neuron). We observed that while the distributions of correlation coefficients are similar, the distributions of p value are not uniform when neurons fire more than two spikes. Since the Benjamini and Hochberg procedure works by ranking the p value from the smallest to largest, such a skewed p value distribution due to spike non-independence (e.g. hippocampal neuronal bursting) can inflate detection of false events. Indeed, even if we now include 20% true events and 80% false events (Author response image 2), around 900 and 1700 false events would be detected as true events at a 5% false discovery rate (using the BH method) in the doublet and triplet condition, respectively. In contrast, a similar number of false events (i.e. 350-450) would be detected in all three conditions as false events, if the α level is lowered on the null distribution (Author response image 1) to reach a 5% empirically estimated false positive rate. We conclude with these analyses that traditional statistical methods may not work as intended with replay sequences, and it is important to find a way to empirically test the consequence of our dataset lacking independent samples. It is important to point out that this observation extends to multiple replay detection methods including those relying on decoding, due to a lack of independence between neighbouring bins in both the temporal and spatial dimension.

**Author response image 1. sa2fig1:** Distribution of Spearman’s rank order correlation score and p value for false events with random sequence where each neuron fires one (left), two (middle) or three (right) spikes.

**Author response image 2. sa2fig2:** Distribution of Spearman’s rank order correlation score and p value for mixture of 20% true events and 80% false events where each neuron fires one (left), two (middle) or three (right) spikes.

**Author response image 3. sa2fig3:** Number of true events (blue) and false events (yellow) detected based on α level 0. 05 (upper left), empirical false positive rate 5% (upper right) and false discovery rate 5% (lower left, based on BH method).

We understand that the reviewers are concerned that our method used to create a randomized dataset could theoretically influence the empirically estimated false positive rates and therefore α level threshold. To examine the effect of data randomization on detection, we created two new types of randomized datasets – a place field circular shifted dataset and a cross experiment cell-id randomized dataset. Place field circular shifted dataset was created in same way as described in the original manuscript, where it was used to create shuffled distributions during replay detection. Its purpose is to validate this concern, using a poor method of randomization (that is identical to the shuffle used in replay detection). The cross experiment cell-id randomized dataset was created by randomly selecting place cells from another experiment, eliminating any potential interaction between the randomization used for creating a sequence, and the shuffle distributions used for detection. We view this as an alternative and robust method of data randomization, which we can then compare to our cell-id shuffled approach used in our original manuscript.

We next performed replay detection using a weighted correlation with a single shuffle method (similar to Figure 3 in our original manuscript), but now applied to these two new randomized datasets. The shuffle methods tested included a time bin permutation shuffle, a spike train circular shift shuffle, a place field circular shift shuffle, or a place bin circular shift shuffle (Author response image 4). Indeed, using a place field shuffle on a place field randomized dataset leads to a 5% empirically estimated false positive rate due the null distribution of the randomized data being too similar to that used for detection. However, the cross experiment and within experiment cell-id randomisation leads to a very similar false positive rate (for each of the other four shuffles used). Interestingly, the estimated false positive rate for the place field randomized dataset overlaps with both cell-id randomized datasets when using shuffles that disrupt only temporal information (i.e. spike train shuffle and time bin shuffle). Together these results suggest that a cell-id based randomization is sufficiently independent from both place-based and temporally-based shuffles for replay detection.

Even though a cross experiment cell-id randomization would be in theory the most independent method of creating a null dataset, its result was extremely similar to that produced by a within experiment cell-id randomization. Given that it is less widely applicable (i.e. it needs multiple recording sessions for it to work) and may not necessarily capture the replay data’s distribution being analyzed, we decided to stay with our original within experiment cell id randomization method to empirically estimate false positive rates. Therefore, based on the additional analyses performed, we would like the reviewers to reconsider their decision about the validity of using a cell-id randomization, as we strongly feel this is an essential part of the study. If we do not account for the estimated false positive rate, we will unable to compare different replay detection methods. A prime example is the use of rank-order correlation to detect events, using either the median spike time or all spikes. On the surface, these two methods appear to have a trade-off – all spikes will detect more events but with a lower mean track discriminability. However, this is simply due to a higher false positive rate, which, when accounted for, show the two methods are very similar (Figure 5C,D). We are happy to include all the analyses in this rebuttal letter as supplementary figures in the manuscript if requested by the reviewers.

**Author response image 4. sa2fig4:** Proportion of false events detected when using dataset with within and cross experiment cell-id randomization and place field randomization. The detection was based on single shuffle including time bin permutation shuffle, spike train circular shift shuffle, place field circular shift shuffle, and place bin circular shift shuffle.

(1b) There was an opinion that "it might still be a useful tool to compare popular replay scoring methods, as long as (1) instances where the threshold is incorrectly referred to as an "adjusted p-value" are removed and (2) it is not recommended as a novel replay detection approach but simply a tool to compare the methods.

We understand that an ‘adjusted p-value’ is a term specifically reserved for multiple comparisons correction, which can confuse the readers about the kind of ‘adjustment’ we are performing. Therefore, as recommended by the reviewers, we have changed in the revised manuscript two terms- (1) ‘p-value threshold’ to ‘α level’ and (2) ‘adjusted’ to ‘FPR-matched’, to be more explicit about determining α level with matched false positive rates (0.05). We hope we have made a sufficient argument in our previous section for the necessity of including FPR-matched values. For the second point, we would like to clarify that it is not our intention to present this tool as a novel replay detection approach. It is indeed merely a novel tool for evaluating different replay detection methods or measuring the “quality” of a replay event (after more traditional sequence detection methods are employed). We have now modified the manuscript in the intro and discussion to make this point more explicit.

2) However, the reviewers felt that there was value in comparing track discriminability vs. sequence detection, particularly for analysis and comparisons including the PRE-RUN data. Reviewers were moderately enthusiastic about an alternative presentation that concentrated on reporting effect sizes rather than p-values alone, and shifted towards understanding the relationships between the factors.

We would like to thank the editors and reviewers for valuing the potential of our work. However, we are confused about where we are being requested to report effect size, and seek further clarification. While the effect size would be useful for comparing two groups, the p-value threshold (now referred to as α level) is for the selection of single replay events, based on a comparison of their replay score and the distribution of replay scores from shuffled events. For each method, we are reporting the estimated false-positive rate, log odds difference and the proportion of events detected at different α levels. To compare these values between methods, we are performing bootstrapping (1000 times with replacement) to calculate the 95% confidence interval of the distribution at each α level (however we are not measuring statistically significant differences between methods and obtaining a p-value).

While it is interesting to further explore how other factors such as behaviour, place field properties and linearity of the replayed trajectory can influence replay detection, it is beyond the scope of this manuscript which has focused on unresolved methodological debates. Our main aim is to introduce this tool to researchers studying replay, to allow the collective exploration of each factor’s contribution to replay detection.

3) Towards this end, in the discussion, the reviewers were also concerned about the idea of restricting analyses only to those neurons that had fields in both environments. This makes sense as a potential way to avoid concerns about spike sorting, but a subsequent paper should include the obviously functionally important population of neurons that have place fields in only one environment.

We would like to thank the reviewers for this suggestion, and may need to further clarify our reasons for using this approach. While sequence detection used all place fields (active on only one track or on both tracks), track discrimination required a restriction to only those neurons that had place fields in both environments. This was needed to avoid any potential bias in track discrimination arising when neurons with single-track fields were included. Every time a neuron with a place field on only one track participates in a detected replay sequence, it will increase the summed posterior probabilities for the decoded replayed track (but not the other track), automatically improving the track discriminability. Note that decoding was normalized across both tracks (so at any time point, the posterior probabilities for each place bin across both tracks summed to 1). Our goal was to make track discriminability as independent as possible from sequence detection, such that the fidelity of the sequence did not impact our assessment of how well a downstream brain region could determine whether track 1 or track 2 was reactivating.

It is worth noting that track discriminability has other potential uses, and in some cases it is desirable to use all the place fields to maximize the discriminability score. However, this is beyond the scope of this study, but something currently in development in the lab.

Reviewer #1 (Recommendations for the authors):1. I would suggest showing the analysis on a single environment and three environments if possible for the log odds measure.

Unfortunately, our current method is optimized for a two track comparison, because Bayesian decoding must be normalized across tracks to calculate track discriminability. While the quantification of log odds on a single environment with different subregions or multiple environments may be possible with modification, it is beyond the scope of this study. Three environments is possible by taking the ratio of one environment compared to the other two (summed), however chance level would no longer be a log odds of zero.

2. It would be ideal for the functions used for the main analysis to be clearly documented in the code repository. At the very least highlighted in the README for use by others.

We have now updated the README file to provide basic information about the codes used for main analysis. We have also provided a simplified version of code for calculating log odds, which can be easily customised by other users.

3. Do these results hold over animals? It would be important to see the variability in conclusions if possible.

We agree with the reviewer that it is important to demonstrate if the results hold for individual sessions. Therefore, we ran the log odds analysis for individual sessions using weighted correlation with two shuffles (place field shuffle and time bin shuffle) or Spearman correlation with inclusion of only median spike. Consistent with our original findings, we observed that replay events from different sessions but the same behavioural state (PRE, RUN or POST) would tend to cluster together especially when the α level with matching false positive rates were used (Author response image 5). We are including subplots A-B from Figure E (without PRE) as Figure 1—figure supplement 2 for Figure 1 in the manuscript to demonstrate that our main findings hold over sessions.

**Author response image 5. sa2fig5:** Mean log odds and proportion of significant events at α level 0. 05 or FDR-matched α level for individual sessions when replay events were detected using (A-B) weighted correlation with two shuffles and (C-D) Spearman correlation using only each cell’s median spike. Different color is used to indicate different behavioural states: PRE (black), RUN (blue) and POST (orange). Different symbol is used to indicate different sessions..

4. Would the conclusions about p-values hold if you applied a multiple comparisons correction like the Benjamini and Hochberg procedure or Storey's q-value first? I would suggest clarifying why the empirically determined adjusted p-value works better than the other multiple comparison measures.

Please refer to our earlier responses to this issue where we provided evidence for why we think empirically determined false-positive-rate-matched α level are more aligned to our goals than a multiple comparisons analysis with a BH procedure.

5. I would clarify that the p-value threshold and significant events are determined by the replay score only in the text.

Thank you for your suggestion. We have now made this point more explicit throughout the manuscript.

6. "computing the summed posteriors (across time bins) within the replay event for each track" – Please clarify or correct this line because I believe the posterior is summed across time and position.

Thank you for your suggestion. Yes, the summed posterior probabilities were summed across time and position within the replay event for each track. We have corrected this line in the manuscript on page 3.

“Sequenceless decoding involved three steps: 1) computing the summed posteriors (across time and space) within the replay event for each track, 2)….”

7. "difference in mean log odds between track 1 and track 2 replay events" – It is unclear what this mean is over. Session? Over all animals?

The mean log odds for each track’s replay events is averaged across all 10 session. We have updated the manuscript to make this information more explicit on page 3:

“To use sequenceless decoding to evaluate replay events, we computed the difference in mean log odds between track 1 and track 2 replay events across all sessions,…”

8. " ripple power threshold is not necessary for RUN replay, but should be applied to POST replay events" – It is my understanding that a multiunit threshold is still applied. It would be important to clarify this if this is the case.

Yes, we agree that this is an important point to clarify. The multiunit threshold is always applied to select the candidate events before applying ripple power threshold. This was previously mentioned on page 7 (see below), but we will reiterate this point in the discussion for additional clarity:

“We next applied our replay analysis framework to test how a stricter criterion for ripple power affects trajectory discriminability. Candidate events were selected based on both elevated MUA (z-score>3) and ripple power limited to a specific range, measured in SD above baseline (i.e. a z-score of 0-3, 3-5, 5-10, or >10).”

9. "a more positive mean log odds difference would indicate a higher trajectory discriminability in the replay content, and a higher confidence in replay quality. In contrast, a mean log odds difference of 0 would suggest that the quality of replay events is comparable to chance level trajectory discriminability, and that the detected event population most likely consists of low fidelity events, indistinguishable from false-positives,"- is there an absolute value being taken here? The measure can be both positive and negative, so unless the track selected by the replay score is always the positive going track, this cannot be quite correct. Please clarify if that is the case and note that the statistic can be negative.

We believe the confusion is arising from using the mean log odds *difference* (not log odds). To summarise our method, each candidate replay event is decoded against track 1 and track 2 template to give replay scores for both templates. Depending on the α level, if the event is significant for track 1, then it is classified as track 1 and vice versa for track 2. The log odds for each event is calculated by taking the log of the ratio between the summed posteriors for track 1 and track 2 (always T1 over T2). The z-score log odds is then quantified based on a distribution of log odds computed by a track ID shuffle. As a result, a more positive z-scored log odds would indicate a greater likelihood of track 1 reactivation whereas a more negative value would indicate a greater likelihood of track 2 reactivation. If replay detection method can correctly identify track 1 events and track 2 events, then the mean log odds value across all track 1 events should be positive and mean log odds value across all track 2 events should be negative. Therefore the mean T1-T2 log odds difference should therefore be positive if the events are correctly detected. If the detection method is poor (many falsely classified events) or if there are no real replay events, then log odds would become noisy and random, which would lead to near zero mean log odds difference. This was indeed observed for the cell-id randomised datasets in our analysis. If the mean log-odds difference is negative, this would indicate that most track 1 events selected according to a sequence-based approach are consistently having higher probability bias towards track 2 and vice versa. While this mismatch could occur for a single event, this should never occur across large sample of replay events (from real data or a randomized dataset).

10. The term "cross-validation" is being used improperly. Cross-validation has a specific meaning in statistics, implying a test and training set. I would suggest using validation, or orthogonal validation instead.

Thank you for your feedback. We also agree that cross-validation can be misleading to the readers. We have now replaced it with ‘cross-checked’ instead in the revised manuscript.

11. I would add individual examples of the log odds procedure and z-score (as in Tirole et al. 2022, Figure 4B) to help the readers understand the procedure better.

We agree with the reviewer that adding individual examples of log odds and sequenceness p value would help the readers to understand our framework better. We have added four individual examples with different level of sequence fidelity and reactivation bias in Figure 1—figure supplement 1.

12. "Based on the desire to maximize the statistical independence of data, we avoided any smoothing in our analysis, as this could increase the false-positive rate of replay detection. Because we did not use spatial or temporal smoothing, we opted to use larger bin sizes for Bayesian decoding (10 cm by 20 ms)." – Using bins of any kind is a form of "smoothing" and choice of bin edges will lead to different answers. Furthermore smoothing is not inherently bad as it can reduce variance and potentially lead to more statistical power. Choosing not to smooth with a gaussian is fine as a choice but I would change this sentence.

We agree with the reviewer that binning is also a form of smoothing and our original sentence could be misleading. However, we would like to point out that use of gaussian smoothing, especially when applied on posterior probability matrix, can cause neighbouring bins to share information, which may further increase the estimated false-positive rate (e.g. especially when a shuffle distribution is created from the smoothed posterior probability). We have modified the sentence to avoid the impression that we think binning is a better smoothing method than smoothing using a filter while still pointing out how different smoothing methods may influence false-positive rate of detection.

“For instance, we did not examine the impact of spatial smoothing (place fields) and temporal smoothing (posterior probability matrix and/or spike train) on replay detection. For this study, we avoided any gaussian smoothing of the posterior probability matrix in our analysis but opted to use relatively large bin sizes for Bayesian decoding (10 cm by 20 ms). It is possible that with smaller bin sizes, replay detection could improve, or yield a different outcome when comparing methodologies. However, as the size of the decoding bin decreases, noise in the decoded trajectory will likely increase or alternatively must be removed using another smoothing method, which in turn may change the false-positive rate of detection..”

13. "To determine if the decrease in detection rate was associated with decrease in the false-positive rate for this detection method, we calculated the false-positive rate (mean fraction of spurious replay events detected across both tracks)" – I would suggest clarifying that the spurious replay events detected are from the shuffled data. It is clear in the figure legend, but not in the main body of the text.

Thank you for your suggestions. We have updated the manuscript accordingly.

“To determine if this decrease in the detection rate was also associated with a decrease in the false-positive rate, we empirically measured the mean fraction of spurious replay events detected across both tracks after the dataset was randomized (by permuting the cell-id of each place cell) as a proxy for false-positive rate.

Reviewer #2 (Recommendations for the authors):I would recommend the authors focus on the analysis of preplay in their dataset. They can better analyze their observation that no reactivation is evident in events with high replay scores, which will be of interest to the field. I believe that the rest of the work is highly problematic.

We would like to thank the reviewer for valuing our analysis about preplay. It is not our goal to single out a single debate in the field, and while this may be one of the more divisive ones, there is also sufficient diversity how candidate replay events are selected (ripple and/or multi-unit activity) and scored (line fitting or weighted correlation). Even if two methods are valid, it is important to understand how their use may impact the analysis.

We also strongly feel that it is necessary to use an independent metric like log odds to validate and compare the performance of different replay methods to understand the amount of discrepancy across methods in terms of track discriminability, proportion of significant events detected and mean estimated false positive rates. For more information, please refer to our response to the public review and essential revisions where we provided a detailed discussion about our choice of α level adjustment based on matching false positive rates empirically estimated using cell-id randomization.

Reviewer #3 (Recommendations for the authors):I have two fundamental suggestions.1) While I agree that the sequenceless decoding is important to cross-validate the main findings of the paper, reading the paper for the first time gave me the (false) impression that the proposed framework of detecting replay somehow uses this measure to select the final replay events. After reading the methods multiple times, it seems to me that first impression was untrue, and the proposed replay detection only depends on adjusting the p-value based on the empirical false positive estimate using a cell-ID shuffle. If this is correct, then the sequenceless decoding only provides an x-axis as additional evidence to help convince the reader of the conclusions of the comparisons.The way the paper is written, I think most readers would make the same initial assumption as I did. The first time false positive estimates are mentioned in the results of the paper, we are directed to Figure 1D which describes the sequenceless cross-validation. Moreover, given the emphasis of this sequenceless approach in the abstract and the entire manuscript, it is easy to conclude that the false-positive rate is estimated as the number of events that don't show sequenceless log-odds higher than the shuffle.This is problematic because many people in the field might dismiss the findings by thinking "Well that's nice but I don't have two tracks, so I can't do any of that". The only place in the manuscript that made me dig deeper was in the discussion, where the authors state "As such, it is critical for replay studies to independently verify and report this false-positive rate even for the experiments involving only one spatial context." Only after reaching this sentence did I even consider that estimating the false positive rate might not require two tracks, and it took quite some time to make sure it doesn't, and yet even now a part of me is still unsure that I got it right. I would urge the authors to make it more clear that the false positive rate can be estimated readily using a simple shuffle and does not require having two tracks.It is my opinion that estimating the false positive rate and taking it into account is very powerful and can be a game changer for replay detection, but it won't happen if people don't understand it was done and how it was done.

We would like to than the reviewer for this suggestion. We have rewritten the manuscript for to clarify our methods, including emphasizing the fact that the empirically estimated false positive rates were derived by running the same sequence-based detection on cell-id randomized dataset rather than based on log odds metric. The reviewer is correct that this process can be done without the requirement of having two tracks.

2) In my initial reading of the paper, I was confused by Figure 4D (and all the figures like it). It sounds counter-intuitive that a more strict method (e.g. 2 shuffles vs 1 shuffle) can end up detecting *more* replay events. It is not trivial point to grasp, and it is easy for readers to get lost here (e.g. the y-axis "proportion of significant events" may be considered by some confused readers to mean some measure of specificity like "among the events selected by each method, how many of them were true positives"). I think something can be added to this section to better guide readers and explain what is being proposed.I'm thinking of something like "Before controlling for the false positive rates associated with each method, as expected, stricter methods found fewer replay events than more permissive methods: for example, a single shuffle selected 50% of awake events as significant, as opposed to 40% that the stricter 2-shuffles method selected (Figure 4C). However, after adjusting the p-value to control for the false positive rate, we found that stricter methods actually resulted in detecting more replay events: the single shuffle awake replay adjusted p-value was 0.007 which brought down the proportion of significant events to 32%, while the stricter 2-shuffles awake replay adjusted p-value was 0.032, resulting in 35% of events detected as significant. This shows that controlling for the empirically measured false positive rate can, almost paradoxically, result in stricter methods detecting more replay events." The "Place+jump distance" is somewhat separate and can be presented after taking the time to explain this very critical point.

This is an excellent suggestion, and we have modified the manuscript based on this feedback (page 12, and see below), and explain that while more shuffle methods will decrease the number of detected replay events, they also lower the false positive rate by an even larger amount. The consequence of this is that when the false positive rate made similar between methods by reducing the α level of the null distribution, the use of additional shuffle types will result in more detected replay events.

“Before controlling for the estimated false positive rates associated with each method, as expected, the stricter methods would detect fewer replay events than the more permissive methods. For instance, at α level of 0.05, around 50% of candidate events would be detected as significant events when using single shuffle, as opposed to 40% when using two shuffles (Figure 4C). However, after adjusting the α level to matching false positive rate of 0.05, we found that stricter methods would detect more replay events. For example, at FPR-matched α level, the proportion of significant events would reduce to 32% for single shuffle (α=0.007) but 35% for the stricter two-shuffle method (α=0.032). This shows that controlling for the empirically measured false positive rate can, almost paradoxically, result in stricter methods with additional shuffle types detecting more replay events.”

[Editors’ note: what follows is the authors’ response to the second round of review.]

The manuscript has been improved but there are some remaining issues that need to be addressed, as outlined below:The reviewers reached a consensus that the paper requires minor revisions. Please see individual reviewer comments below. In particular, there was a consensus that the discussion of the paper needs to clearly articulate a proper interpretation of the corrected p-values. Specifically, that using one shuffle (cell-ID) to correct the p-values of another (i.e., circular) on a collection of events is not the same as somehow correcting the replay detection for individual events.Reviewer #1 (Recommendations for the authors):Thank you to the authors for their thoughtful responses to the reviews. I do think the response clarified several aspects of the manuscript and improved the clarity of the manuscript overall. The documentation and improvement of the code associated with the manuscript is also improved.As I understand the logic, the main claims of the framework to study sequence detection methods are:1. A good sequence detection method should assign the same track as track decoding.2. The significant replay events for a given method can be adjusted by matching the false positive rate resulting in greater track discriminability (and more conservatively identified replay events)3. This can be used to compare replay detection methods or quality of replay eventThe strengths of the work are:1. it is more conservative in judging which sequence is a replay or "spurious" given that the two track model is true2. it makes some interesting conclusions about the use of the ripple power and multiunit as criterion for detecting replay events.

We would like to thank the reviewer for the constructive feedback and comments, and highlighting the value in our framework for comparing replay detection methods.

The weaknesses of the work are:1. The framework relies on an adjustment for track discriminability but, as the authors acknowledge, there are many ways in which the model can be misspecified. This only tests for a very specific way in which they are misspecified. And it doesn't seem to give one much insight into why a particular method is better. For example, the authors' response seems concerned about the lack of accounting for bursting in replay detection, but track discriminability does not circumvent this. The way to account for lack of independence in time bins is to explicitly account for the bursting, such as fitting the place field estimates with a self-history term. The track discriminability measure simply marginalizes over the time and position and already incorporates the misspecified encoding model.

The log odds for each event quantified the relative probabilistic bias towards track 1 vs track 2 independently of the temporal fidelity of the replay sequence. While we relied on Track discriminability (computed from the log odds) to quantify how well a detected replay sequence was track-specific in its reactivation, we did not use log odds per se for any kind of adjustment or replay event selection. Track discriminability was only quantified using a distribution of replay events, with the expectation that the log odds separation between track 1+2 replay events increases with a better detection of track-specific replay events. This provided an opportunity to see how the log odds difference changed as a function of detection parameter such as ripple power, shuffle types and so on.

We adjusted the α level between replay detection methods, such that a similar proportion of false-positive events were obtained between methods, to compare the detection rate and track discriminability in a more equitable manner. Without doing this, stricter replay detection methods are expected to have a high track discriminability but lower detection rate; by matching the false positive rate, this “strictness” is matched in at least one respect. Nevertheless, even without any adjustment based on FPR, most findings comparing the mean log odds difference of different methods or behavioural states at an α level of 0.05 would still hold. This suggests that our results were not attributable to a specific method of adjustment.

We agree with the reviewer that one of the many factors that we were interested in was the impact of neuronal bursting on inflating the false positive rate of neuronal sequence detection. This was explored by comparing the detection performance using a rank-order correlation when either all spike times or only the median spike time from each neuron was included in the analysis. We found a significant increase in the false positive rate when including all spikes (presumably due to bursting). However, more generally for replay detection methods using decoding, it is the fact that correlations exist in the decoded bins for both the space and time dimension, and while the hope is that performing shuffles in these two dimension circumvents the problems of performing a statistical measure (e.g. weighted correlation) without having independent bins, we lack a ground truth to directly validate this assumption. While our log-odds metric was not used for replay detection, it is important to point out that it is less affected by neuronal bursting, as it is not measuring the “sequenceness” of a replay event.

We agree that there may be other possible methods (in addition to log odds) for measuring the “quality” of replay events. It is important to point out that our use of log-odds was not to optimize detection, but cross check events, to see which method of replay sequence detection was more optimal when comparing distributions of replay events detected.

2. This framework relies on classifying aspects of the replay event as significant or not significant at a given significance threshold. This is reliant on the particular null distribution specified.

We agree with the reviewer that replay sequence detection still relied on comparing sequence score of a given candidate relative to one or multiple null distribution specified. This is because this study was meant to evaluate existing replay detection methods (which all relied on this statistical approach) rather than designing novel approaches. Furthermore, to capture the performance of each method more comprehensively, we also showed the log odds difference and detection rate using a range of α levels (0.2 to 0.001).

3. The generality of the work is limited by their two track framing.

We agree with the reviewer that this work relies on a two track framework, which is less commonly used in replay studies than a single track. However, we do believe that differences between replay methods quantified here would hold for other studies regardless of whether two tracks or only a single track is used. It is worth noting that in the cases when only one track is used, it may be possible to quantify log odds by creating a virtual second track based on recorded place cell statistics or by comparing the discriminability across different portions/arms of the maze (depending on the experimental design), although this is beyond the scope of our current study. We have briefly discussed this on page 17 (discussion) in the revised manuscript.

The work claims that their framework is a "unifying approach to quantify and compare the replay detection performance". The work falls short of doing this in general but I do think it accomplishes this with a much narrower scope, namely in adjusting methods that fail to discriminate between tracks appropriately in the case of two tracks.

We agree with the reviewer that this sentence could be misleading and not well-phrased. We have removed the phrase “unifying framework”, and updated the manuscript to reflect the fact that this framework was accomplishing replay detection comparison within a narrower scope.

– Are the significant events being corrected for multiple comparisons? Could the authors explain why no corrections are needed?

While the likelihood of detecting any significant replay event increases with an additional track, our detection of replay (compared to a null distribution) is performed separately for each track. If each track forms an independent hypothesis (e.g. is this a track 1 replay event), there is only a single comparison being made, and no correction for multiple comparisons is needed.

– I do not think the authors sufficiently addressed the question of what happens if the track discrimination model is misspecified in their response. For example, if there are truly three tracks but you only account for two?

We agree with the reviewer that there are several assumptions about the track discrimination framework.

Firstly, it is assumed that only one track was usually replayed during a candidate ripple event, which might not be always true. However, based on our dataset, most methods would detect a rather low proportion of events being classified significant for both tracks. In addition, the proportion of multitrack events (out of all significant events at a given α level such as 0.05) were similar across PRE, RUN and POST and were comparable to the proportion of empirically measured false positive events obtained from a randomised dataset (see Figure 6- supplementary figure 6 in the revised manuscript). This suggests that most multi-track events we observed were probably mostly due to false positive detection (on one of the two tracks) rather than multiple track representations in a single replay event.

Furthermore, prior experience on other tracks not analysed in the log odds calculation should not pose any issue, given that the animal likely replays many experiences of the day (e.g. the homecage or rest pot). These “other” replay events likely contribute to candidate replay events that fail to have a statistically significant replay score on either track. Furthermore, even when the replay events of other track experiences were falsely detected based the sequential content, given that the representations of other experiences were sufficiently different from both tracks we analysed, the log odds measure for these events should be close to 0 as the representations should not be more similar to track 1 or track 2.

Reviewer #2 (Recommendations for the authors):While I appreciate the conceptual novelty of using the log odds ratio as a metric for testing the noisiness of replay events and comparing this metric with results from sequence analyses, unfortunately, I was not persuaded by the arguments presented in the authors' rebuttal. I think the study would benefit greatly if the authors consulted or collaborated closely with an expert on statistical methods, particularly on the application of resampling methods, to enhance the rigor of their approach. Without this, I remain concerned that this work can cause further confusion (and noise) rather than clarity in these analyses.I believe I noted my concerns to great length in the initial review, but perhaps the authors will wish to look through the section "The problem with replay decoding: shuffles" section in Foster 2017 and note that each shuffle is permissive for some events and not others. E.g. if the circular shuffle is likely to pass through events that arise from edge effects, their study does not explain how a shift of the α level is going to fix that issue and remove false positives. Each type of shuffle will pass through different subsets of events, some of which may be "false positives," but they will be different events in each case. Simply adjusting the α levels or p-values, the solution proposed by this study, will not fix the underlying reasons for the false positives, and lead to potentially mistaken confidence in the results.

We would like to thank the reviewer for the feedback and comment.

Firstly, we would like the reviewer for appreciating the novelty and potential usefulness of this crosschecking framework in evaluating replay under the constraints of having no ground truth for validation.

First, to address the concern about the use of an empirically estimated false positive rate based on cell Id randomization, we agree with the reviewer that an α level adjustment is not a quick fix to a poor method generally (although rank-order correlation did appear to be an exception to this). The most effective solution is using multiple, orthogonally-designed shuffles that directly address possible correlations in space and time found in place fields and spike trains (respectively). We have made this point clearer in our discussion, which is supported by our observation that decoding methods that use multiple shuffle types always perform better than an α-level adjusted single shuffle type method.

An ideal method will minimise false-positives and maximise true events, and by changing the α level threshold, you are reducing both, but not necessarily by the same amount (hopefully reducing false positives more than true events). But without matching the false-positive rate, it is impossible to directly compare methods.

Indeed, one of the main concerns about use of a cell-id randomized dataset is that it may be underestimating the false positive rate in a way that is biased towards specific shuffling distributions (I.e. place-based or time-based shuffling). In the original rebuttal letter, we discussed how use of cell id randomized dataset would be preferred over a place field randomized dataset as a cell id randomized dataset was not underestimating empirically-estimated false positive rates compared to place field randomized dataset. Here we extend the analysis to demonstrate that, in most cases, a cell id randomized dataset was also not underestimating empirically-estimated false positive rates compared to spike train shuffled dataset (Figure B). In particular, using shuffling procedures that disrupt place or temporal information causes a substantial reduction in the empirically-estimated false positive rates when they were applied to place field randomized dataset or spike train randomized dataset, respectively. In contrast, for most cases, a cell id randomized dataset would lead to highest empirically-estimated false positive rates, suggesting that, while imperfect, cell id randomization is the relatively more conservative null distribution for estimating false positive rates.

We have updated the manuscript to discuss further about the purpose and caveats/limitation of this FPR-based adjustment to avoid confusion and over-confidence about this false-positive estimates among the readers. We have also included Figure 1 – supplement 2 in the results and discussion of our manuscript

We have consulted with a statistician (Dr. Henrik Singmann) for this work, which is now noted in the acknowledgements.

Reviewer #3 (Recommendations for the authors):I find that the revised manuscript has considerably improved. I have one outstanding issue with the randomization shuffle, which can actually be alleviated if the authors update Figure D to include the 3 remaining shuffles, and a minor comment.Outstanding question regarding the randomization shuffle:My concerns about the cell-id shuffle were that (1) any shuffle, including this one, may actually underestimate the true false positive rate, and (2) this tendency to underestimate the true false positive rate will be exacerbated when the randomization shuffle is similar to the shuffle used for detecting replay events, resulting in a biased comparison between methods.I think (1) is unavoidable as it would be hard to disprove any structure that could be inadvertently erased by any given shuffle. The true false positive rate may always be higher than our best estimates captured by our surrogate datasets, and our estimates are simply a lower bound of the true false positive rate. In particular for the cell-id shuffle (including the "cross experiment cell-id" shuffle), there may be some underlying structure in biological data which violates assumptions of independence that is not captured by place-field swapping. This would undoubtedly be the case when including a variety of cell types with generally different place field properties (e.g. along the dorsoventral axis or the deep/superficial sublayer). Even without different anatomical cell types, more excitable cells may be more likely to have multiple place fields as well as firing more spikes within candidate events – a potential structure that is not captured by the cell-id shuffle.While concern (1) only needs to be mentioned (in the discussion or in a public review), concern (2) is particularly problematic because the authors compare the (FRP-matched) performance of different methods to conclude which are best. Figure D shows a comparison of two methods: place field shifting and place field swapping (with place fields within the same dataset, the randomization used throughout the manuscript, or from a different dataset, which is almost identical to the original randomization) before trajectory decoding. Figure D convincingly argues that relative to field swapping (cell-id shuffle), field shifting (place field circular shifted shuffle) is a much poorer method of randomization as it considerably underestimates the false positive rate for detection methods using similar shuffles. Note that the bias is slightly attenuated for the "place bin shuffle" (it is worse for the "place field shuffle"), demonstrating that there is a confounding gradient with larger similarities between the randomization and the detection shuffles resulting in larger bias.According to the authors, Author response image 4 shows that "cell-id based randomization is sufficiently independent from both place-based and temporally-based shuffles for replay detection". I cannot see how that can be concluded from the figure. The place field swapping methods may well be underestimating the false positive rate unevenly. Place field swapping is still a pre-decoding place-based shuffle, so the fact that its estimate agrees with the estimate of another pre-decoding place-based shuffle (place field shifting) does not demonstrate that these estimates are not a biased. I would like to stress that one of the authors' conclusions is that "more shuffles, with a preference for pre-decoding shuffles, lead to better replay detection." All of the approaches using "more shuffles, with a preference for pre-decoding shuffles" included place field shifting, which in my opinion is similar to the place field swapping of the cell-id shuffle.I commend the authors for the approach of Author response image 4, which is a good way to address this key concern. I recommend this figure to be included in the manuscript. Moreover, if the authors were to include the other 3 shuffles (it currently includes "Shuffle 2" from the methods) and show that the cell-id does not underestimate the false positive rate more than those shuffles in a biased way, that would practically disprove this issue. Indeed, the concern is that the cell-id shuffle may be underestimating the false positive rate particularly for detection methods using pre-decoding and place-based shuffles, but if none of the other 3 shuffles produce higher false positive estimates for the lower row in Figure D, that would be convincing evidence that the cell-id randomization procedure is not biased towards detection methods employing pre-decoding place-based shuffles and strengthen the conclusions.

Thank you for your feedback.

[Editors’ note: what follows is the authors’ response to the third round of review.]

The manuscript has been improved but there are some remaining issues that need to be addressed, as outlined below:The introduction of the manuscript has already been significantly revised. However, the final revision requested by the reviewers is to spend a few sentences outlining not only sequence scoring, but also the fact that the manuscript utilises multiple types of surrogate data sets to establish null distributions. In particular, it is suggested that some assumptions about the utility of the cell-id shuffle surrogate be described, as well as the importance of track discriminability. Reviewer #2 (see below) is quite proscriptive; please consider their points and update the introduction to at least mention these issues up front.

We would like to thank the reviewers and editors for their feedback and comments. We agree with the reviewers that it is beneficial to include a section stating our framework’s assumptions including (1) the utility of using a randomized surrogate dataset to empirically estimate the false positive rate and (2) the value of measuring track discriminability for the events detected based on track-specific sequenceness. We have updated our manuscript based on the feedback here and feedback from Reviewer #2 (please see our detailed responses to Reviewer #2).

In particular, we included three main assumptions associated with this framework during introduction section.

Firstly, we stated that ‘for a given replay detection method, the proportion of spurious events (generated from randomized data) that are detected as significant events can be used to empirically estimate the proportion of non-replay events that are falsely labelled as significant replay events in real hippocampal data. However, this assumes that the null distribution used for creating randomized data is not used for detection, and that it does not underestimate the false-positive rates compared to the other null distribution(s) used for detection’. We wanted to highlight the utility of randomized dataset in estimating false positive rates with the caveat that this same method should not be used to create a null-distribution for replay detection and should not underestimate the false-positive rates compared to other null distributions. We hoped to introduce what a preferred null distribution for creating randomized dataset should look like without going too much into the specifics of shuffling procedures to early in the manuscript, which would detract from its readability. However, during the early part of the Results section when we first described our use of empiricallyestimated false positive rates, we now offer a more detailed description of the four potential null distributions for generating randomized datasets and explained why cell-id shuffling was the preferred choice in our study.

Secondly, we stated that ‘for neural data, pooled from multiple animals, with track-specific replay sequences but not spurious replay events, track discriminability should correlate with the sequenceness score of the track-specific replay’. We wanted to highlight that track discriminability measure should be able to reflect the overall fidelity of the replay events only when these events are detected from neural data that contain replay. When events were detected from a randomized dataset or from a real dataset that contains very few replay events, there should be a dissociation between the sequenceness score and track discriminability.

Lastly, we stated that ‘the empirically-estimated false positive rate can vary between methods, but can be adjusted by a scaling correction of the α level such that the performance of different replay methods can be evaluated and compared at an equivalent empirically-estimated false positive rate. Note that this method of scaling the α level is designed for method comparison only, and should not be meant as a substitute for using appropriate shuffle methods to create a sufficient set of null distributions for detection’. This assumption leads to our approach of applying a scaling correction to an equivalent empirically estimated false-positive rate, enabling more equitable comparisons of various replay detection methods. We wanted to emphasize that this approach was designed to facilitate fair comparison among methods, not to serve as a substitute for additional shuffle techniques in replay detection.

Reviewer #2 (Recommendations for the authors):The revision added some text regarding the limitations of the study. I was hoping that the revised manuscript would also be more explicit regarding the underlying assumptions. While I don't think reviewers should do the authors work for them, after such a lengthy review, perhaps this is the only way short of a rejection.From what I can tell, these are the main assumptions. If I am wrong, perhaps the other reviewers, or the authors can correct me. Also perhaps there are others that I missed.1) After recording from two tracks, events with low discriminability between these two tracks cannot be said to be replays, regardless of their sequence scores.– Pooled across data, discriminability generally correlates with replay score, though this differs across datasets and timepoints of recording.

We agree with reviewer that this framework assumed that detected replay events with low track discriminability are assumed to be false positive events, regardless of their sequence score. While this sequence score can be observed to correlate with track discriminability, this should not occur when using a randomized data set (composed only of false-positive events), or for a real dataset that does not contain replay events.

We have stated this assumption (sequence score correlating with track discriminability for real replay events but not spurious events) in the manuscript’s introduction

2) The cell-id shuffle provides the best null distribution to test against for replay.– This is so because cell-id shuffled events show low discriminability even in instances where they have high sequence scores

We agree with reviewer that this framework assumed that a cell-id randomization can create a randomized dataset to measure the false-positive detection rate of a given replay detection method, with the caveat that this same method is not used to create a null-distribution for replay detection. It is true that cell-id randomized events show low track discriminability even in instances when they have high sequence scores. However, it is not a preferred null distribution for this reason, but rather because it is sufficiently different from the other shuffling methods, and as such does not underestimate the false positive rate.

We believe a discussion about why a particular null distribution is preferred over other null distributions during the early part of the introduction can distract the reader from the main points of the paper, and have moved this to the Results section. However, in the introduction section, we have decided to include our assumption that using a randomized surrogate dataset is useful for empirically estimating the false positive rate. This is based on the premise that the null distribution used to create the randomized dataset is not used for detection, and thus does not underestimate false positive rates compared to other null distributions used in detection. Furthermore, as mentioned in our response to the reviewer's first point, during the introduction, we have decided to introduce our assumption about how the sequence score would correlate with track discriminability for real replay events but not spurious events.

3) The proportion of cell-id shuffled events that qualify as replays in any given period is similar to the proportion of events generated in the real data from hippocampus network that are likewise not actual replay but are labeled as significantly ordered.– This therefore allows for a scaling correction to bring this proportion in line with the α level.

We agree with reviewer that this framework assumed that the proportion of cell-id randomised events that were falsely detected as significant events would be similar to the proportion of nonreplay events that were falsely labelled as significant replay events in real hippocampal data. This assumption leads to our approach of applying a scaling correction to create an equivalent empirically-estimated false-positive rate, which would allow us to compare the performance of various replay detection methods. We want to emphasize that this approach is for comparing replay detection methods more equitably, but is not meant to be used as an alternative to using additional shuffle methods when performing replay detection.

We have included this assumption in the introduction section of the manuscript.

4) In the absence of performing cell-id shuffles, which is straightforward, the alternative prescription is to record from two tracks and adjust the apparent α to match those of cell-id shuffled events obtained during the recording– This is so that the same relative number of events are labeled as replays, even though they may not be the actual replays.

We agree with the reviewer that this framework assumes that we can adjust the α level based on the empirically-estimated false positive rates such that the performance of various methods can be evaluated at a level that leads to similar relative number of false events. However, the adjustment is not designed to fix the detection issues and selectively remove true false positive events, rather it aims to permit an equitable comparison between methods. Furthermore, using a cell-id randomized null distribution does not solve the problem of not knowing equivalent falsepositive rates for each replay detection method, which would render any method comparison difficult.

We have included our assumption about the use of a scaling correction of α level such that the performance of various replay methods can be compared at an equivalent empirically-estimated false positive rate. However, we decided to keep the detailed discussion about the caveats of this approach still in the discussion part of the manuscript rather than move them to the introduction as we felt it would overload the readers too early on with detailed information in an unhelpful way.